# Mapping cell type-specific transcriptional enhancers using high affinity, lineage-specific Ep300 bioChIP-seq

Pingzhu Zhou[1†], Fei Gu[1†], Lina Zhang[2], Brynn N Akerberg[1], Qing Ma[1], Kai Li[1], Aibin He[3], Zhiqiang Lin[1], Sean M Stevens[1], Bin Zhou[4], William T Pu[1,5*]

[1]Department of Cardiology, Boston Children's Hospital, Boston, United States; [2]Department of Biochemistry, Institute of Basic Medicine, Shanghai University of Traditional Chinese Medicine, Shanghai, China; [3]Institute of Molecular Medicine, Peking-Tsinghua Center for Life Sciences, Peking University, Beijing, China; [4]State Key Laboratory of Cell Biology, CAS center for Excellence in Molecular Cell Science, Shanghai Institute of Biochemistry and Cell Biology, Chinese Academy of Sciences, Shanghai, China; [5]Harvard Stem Cell Institute, Harvard University, Cambridge, United States

*For correspondence: wpu@pulab.org

[†]These authors contributed equally to this work

Competing interests: The authors declare that no competing interests exist.

**Abstract** Understanding the mechanisms that regulate cell type-specific transcriptional programs requires developing a lexicon of their genomic regulatory elements. We developed a lineage-selective method to map transcriptional enhancers, regulatory genomic regions that activate transcription, in mice. Since most tissue-specific enhancers are bound by the transcriptional co-activator Ep300, we used Cre-directed, lineage-specific Ep300 biotinylation and pulldown on immobilized streptavidin followed by next generation sequencing of co-precipitated DNA to identify lineage-specific enhancers. By driving this system with lineage-specific Cre transgenes, we mapped enhancers active in embryonic endothelial cells/blood or skeletal muscle. Analysis of these enhancers identified new transcription factor heterodimer motifs that likely regulate transcription in these lineages. Furthermore, we identified candidate enhancers that regulate adult heart- or lung-specific endothelial cell specialization. Our strategy for tissue-specific protein biotinylation opens new avenues for studying lineage-specific protein-DNA and protein-protein interactions.

## Introduction

The diverse cell types of a multicellular organism share the same genome but express distinct gene expression programs. In mammals, precise cell-type specific regulation of gene expression depends on transcriptional enhancers, non-coding regions of the genome required to activate expression of their target genes (*Visel et al., 2009a*; *Bulger and Groudine, 2011*). Enhancers are bound by transcription factors and transcriptional co-activators, which then contact RNA polymerase two engaged at the promoter, stimulating gene transcription.

Enhancers are nodal points of transcriptional networks, integrating multiple upstream signals to regulate gene expression. Because enhancers do not have defined sequence or location with respect to their target genes, mapping enhancers is a major bottleneck for delineating transcriptional networks. Recently chromatin immunoprecipitation of enhancer features followed by sequencing (ChIP-seq) has been used to map potential enhancer. DNase hypersensitivity (*Crawford et al., 2006*; *Thurman et al., 2012*), H3K27ac (histone H3 acetylated on lysine 27) occupancy (*Creyghton et al., 2010*; *Nord et al., 2013*), or H3K4me1 (histone H3 mono-methylated on lysine 4) occupancy (*Heintzman et al., 2007*) are chromatin features that have been used to identify cell-type- specific

enhancers. While most enhancers are DNase hypersensitive, DNase hypersensitive regions are often not active enhancers (*Crawford et al., 2006*; *Thurman et al., 2012*). H3K27ac is enriched on cell type-specific enhancers (*Creyghton et al., 2010*; *Nord et al., 2013*), but may be a less accurate predictor of enhancers than other transcriptional regulators (*Dogan et al., 2015*). Chromatin occupancy of Ep300, a transcriptional co-activator that catalyzes H3K27ac deposition, has been found to accurately predict active enhancers (*Visel et al., 2009b*). However, antibodies for *Ep300* are marginal for robust ChIP-seq, particularly from tissues, leading to low reproducibility, variation between antibody lots, and inefficient enhancer identification (*Gasper et al., 2014*).

Mammalian tissues are composed of multiple cell types, each with their own lineage-specific transcriptional enhancers. Thus defining lineage-specific enhancers from mammalian tissues requires developing strategies that overcome the cellular heterogeneity of mammalian tissues, particularly when the lineage of interest comprises a small fraction of the cells in the tissue. Past efforts to surmount this challenge have taken the strategy of purifying nuclei from the cell type of interest using a lineage-specific tag. For instance, nuclei labeled by lineage-specific expression of a fluorescent protein have been purified by FACS (*Bonn et al., 2012*). This method is limited by the need to dissociate tissues and recover intact nuclei, and by the relatively slow rate of FACS and the need to collect millions of labeled nuclei. To circumvent the FACS bottleneck, cell type-specific overexpression of tagged SUN1, a nuclear envelope protein, has been used to permit affinity purification of nuclei (*Deal and Henikoff, 2010*; *Mo et al., 2015*). Although this mouse line was reported to be normal, SUN1 overexpression potentially could affect cell phenotype and gene regulation (*Chen et al., 2012*). Chromatin from isolated nuclei are then subjected to ChIP-seq to identify histone signatures of enhancer activity. However, as noted above histone signatures may less accurately predict enhancer activity compared to occupancy by key transcriptional regulators (*Dogan et al., 2015*).

Here, we report an approach to identify murine enhancers active in a specific lineage within a tissue. We developed a knock-in allele of *Ep300* in which the protein is labeled by the *bio* peptide sequence (*de Boer et al., 2003*; *He et al., 2011*). Cre recombinase-directed, cell type specific expression of BirA, an E. coli enzyme that biotinylates the *bio* epitope tag (*de Boer et al., 2003*), allows selective *Ep300* ChIP-seq, thereby identifying enhancers active in the cell type of interest. Using this strategy, we identified thousands of endothelial cell (EC) and skeletal muscle lineage enhancers active during embryonic development. Extending the approach to adult organs, we defined adult EC enhancers, including enhancers associated with distinct EC gene expression programs in heart compared to lung. Analysis of motifs enriched in EC or skeletal muscle lineage enhancers predicted novel transcription factor motif signatures that govern EC gene expression.

## Results

### Efficient identification of enhancers using Ep300$^{fb}$ bioChIP-seq

We developed an epitope-tagged *Ep300* allele, *Ep300$^{fb}$*, in which FLAG and *bio* epitopes (*de Boer et al., 2003*; *He et al., 2011*) were knocked into the C-terminus of endogenous *Ep300* (*Figure 1A–B* and *Figure 1—figure supplement 1A*). Transgenically expressed BirA (*Driegen et al., 2005*) biotinylates the *bio* epitope, permitting quantitative Ep300 pull down on streptavidin beads (*Figure 1C*). We have not noted abnormal phenotypes. Heart development and function are sensitive to *Ep300* gene dosage (*Shikama et al., 2003*; *Wei et al., 2008*), yet *Ep300$^{fb/fb}$* homozygous mice survived normally (*Figure 1D*) and *Ep300$^{fb/fb}$* hearts expressed normal levels of Ep300 and had normal size and function (*Figure 1—figure supplement 1B–E*). These data indicate that *Ep300$^{fb}$* is not overtly hypomorphic.

To evaluate *Ep300$^{fb}$*-based mapping of Ep300 chromatin occupancy, we isolated embryonic stem cells (ESCs) from *Ep300$^{fb/fb}$; Rosa26$^{BirA/BirA}$* mice. We then performed Ep300$^{fb}$biotin-mediated chromatin precipitation followed by sequencing (bioChiP-seq), in which high affinity biotin-streptavidin interaction is used to pull down Ep300 and its associated chromatin (*He et al., 2011*). Biological duplicate sample signals and peak calls correlated well (93.6% overlap; Spearman r = 0.96; *Figure 2A–B*). We compared the results to publicly available Ep300 antibody ChIP-seq data generated by ENCODE (overlap between duplicate peaks 77.8%; r = 0.91; *Figure 2A–B*). Ep300 bioChiP-seq identified 48963 Ep300-bound regions ('Ep300 regions') shared by both replicates, compared to 15281 for Ep300 antibody ChIP-seq (*Figure 2A,C*). The large majority (89.6%) of Ep300 regions

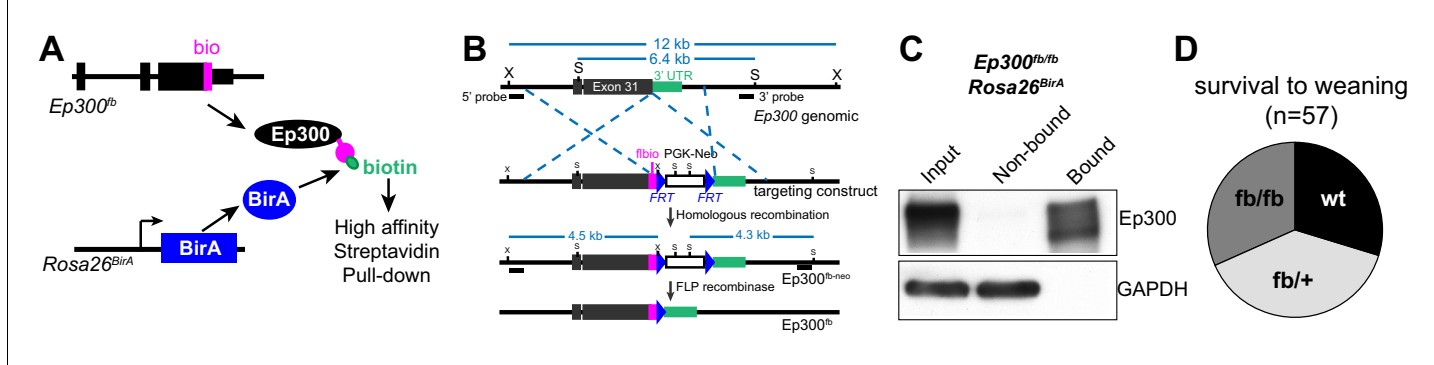

**Figure 1.** Generation and characterization of Ep300flbio allele. (**A**) Experimental strategy for high affinity Ep300 pull down. A flag-bio epitope was knocked onto C-terminus of the endogenous Ep300 gene. The *bio* peptide sequence is biotinylated by BirA, widely expressed from the Rosa26 locus. (**B**) Targeting strategy to knock flag-bio epitope into the C-terminus of Ep300. A targeting vector containing homology arms, flag-bio epitope, and Frt-neo-Frt cassette was used to insert the epitope tag into embryonic stem cells by homologous recombination. Chimeric mice were mated with Act:Flpe mice to excise the Frt-neo-Frt cassette in the germline, yielding the *Ep300fb* allele. (**C**) Biotinylated Ep300 is quantitatively pulled down by streptavidin beads. Protein extract from *Ep300flbio/flbio*; *Rosa26BirA* embryos was incubated with immobilized streptavidin. Input, bound, and unbound fractions were analyzed for Ep300 by immunoblotting. GAPDH was used as an internal control. (**D**) Ep300flbio/flbio mice from heterozygous intercrosses survived normally to weaning.

The following figure supplement is available for figure 1:

**Figure supplement 1.** Characterization of Ep300fb allele.

detected by antibody were also found by Ep300 bioChiP-seq, and Ep300 signal was substantially stronger using bioChiP-seq (*Figure 2A,C,D*). These data indicate that Ep300fb bioChiP-seq has improved sensitivity compared to Ep300 antibody ChIP-seq for mapping Ep300 chromatin occupancy in cultured cells.

## Identification of tissue-specific enhancers using Ep300fb bioChIP-seq

We used *Ep300fb/+*; *Rosa26BirA/+* mice to analyze Ep300fb genome-wide occupancy in embryonic heart and forebrain. We performed bioChiP-seq on heart and forebrain from embryonic day 12.5 (E12.5) embryos in biological duplicate (*Figure 3A–B*). There was high reproducibility (83% and 93%, respectively) between biological duplicates (*Figure 3B* and *Figure 3—figure supplement 1A*). In comparison, published Ep300 antibody ChIP-seq from E11.5 heart and forebrain (*Visel et al., 2009b*) had lower signal-to-noise and yielded few peaks when analyzed using the same peak detection algorithm (MACS2 [*Zhang et al., 2008*]). Using the originally published peaks, antibody-based Ep300 ChIP-seq yielded 9.5x or 3.0x less Ep300 regions in heart and forebrain, respectively (*Figure 3B*). These regions overlapped 58.7% and 64.7% of the Ep300fb bioChIP-seq regions, suggesting that the epitope-tagged allele has superior sensitivity and specificity for mapping Ep300-bound regions in tissues, as it does in cultured cells.

We compared Ep300fb regions from forebrain and heart (*Supplementary file 1*). Only a minority of Ep300fb regions (8.9% for heart and 31.3% for brain) were common between tissues (*Figure 3A*). Viewing Ep300fb bioChiP-seq signal at genes selectively expressed in heart or brain confirmed robust tissue-specific differences that overlapped enhancers with known tissue-specific activity (*Figure 3—figure supplement 1B*). Genes neighboring the Ep300fb occupied regions specific to heart or forebrain were enriched for gene ontology (GO) functional terms relevant to the respective tissue (*Figure 3C*). These results reinforce the conclusion that Ep300 occupies tissue-specific enhancers and indicate that this conclusion was not a consequence of insensitive detection of Ep300-occupied regions in earlier studies (*Visel et al., 2009b*).

Ep300 is a histone acetyltransferase, and one of its enzymatic products is histone H3 acetylated on lysine 27 (H3K27ac). We compared the genome-wide signal of Ep300fb and H3K27ac in E12.5 heart and forebrain (*Figure 3—figure supplement 1C* and data not shown). There was a high correlation between biological replicates (r = 0.98). Ep300 was also well correlated with H3K27ac

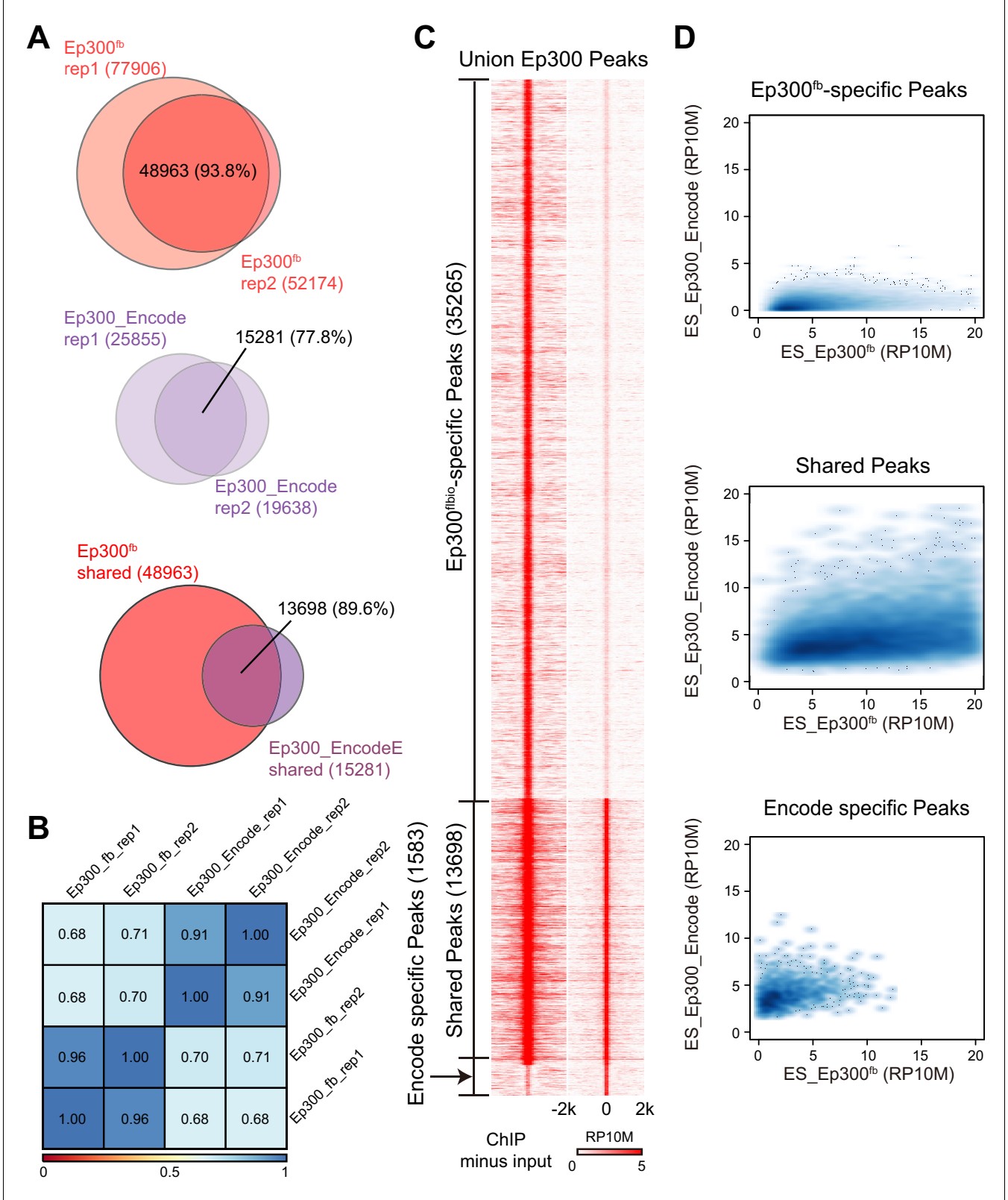

**Figure 2.** Comparison of Ep300 bioChiP-seq to antibody ChIP-seq for mapping Ep300 chromatin occupancy. (**A–B**) Comparison of biological duplicate antibody Ep300 ChIP-seq (Encode) and Ep300 bioChiP-seq (flbio). The Ep300 bioChiP-seq data had greater overlap between replicates and greater intragroup correlation. Most antibody peaks were covered by the bioChiP-seq data. There were 3.3 times more Ep300 regions detected by Ep300 bioChiP-seq. 89.6% of Ep300 regions detected by antibody ChIP-seq were recovered by Ep300 bioChIP. Panel B shows Spearman correlation between

*Figure 2 continued on next page*

*Figure 2 continued*
samples over the peak regions. (C) Tag heatmap shows input-subtracted Ep300 antibody or bioChIP signal in the union of the Ep300-bound regions detected by each method. (D) Correlation plots show greater Ep300 bioChIP-seq signal compared to antibody bioChIP-seq.

(r = 0.64), independently validating the Ep300[fb] bioChIP-seq data. The previously published Ep300 antibody ChIP-seq data (*Visel et al., 2009b*) was less well correlated to H3K27ac (r = 0.37), although the correlation was highly statistically significant (p<0.0001). Interestingly, 26.4% and 52.9% of heart and brain H3K27ac regions were shared between tissues (*Figure 3—figure supplement 1D*) compared to 8.9% and 31.3% for Ep300[fb] heart and brain regions, respectively (*Figure 3A*), suggesting that Ep300[fb] occupancy is more tissue-specific.

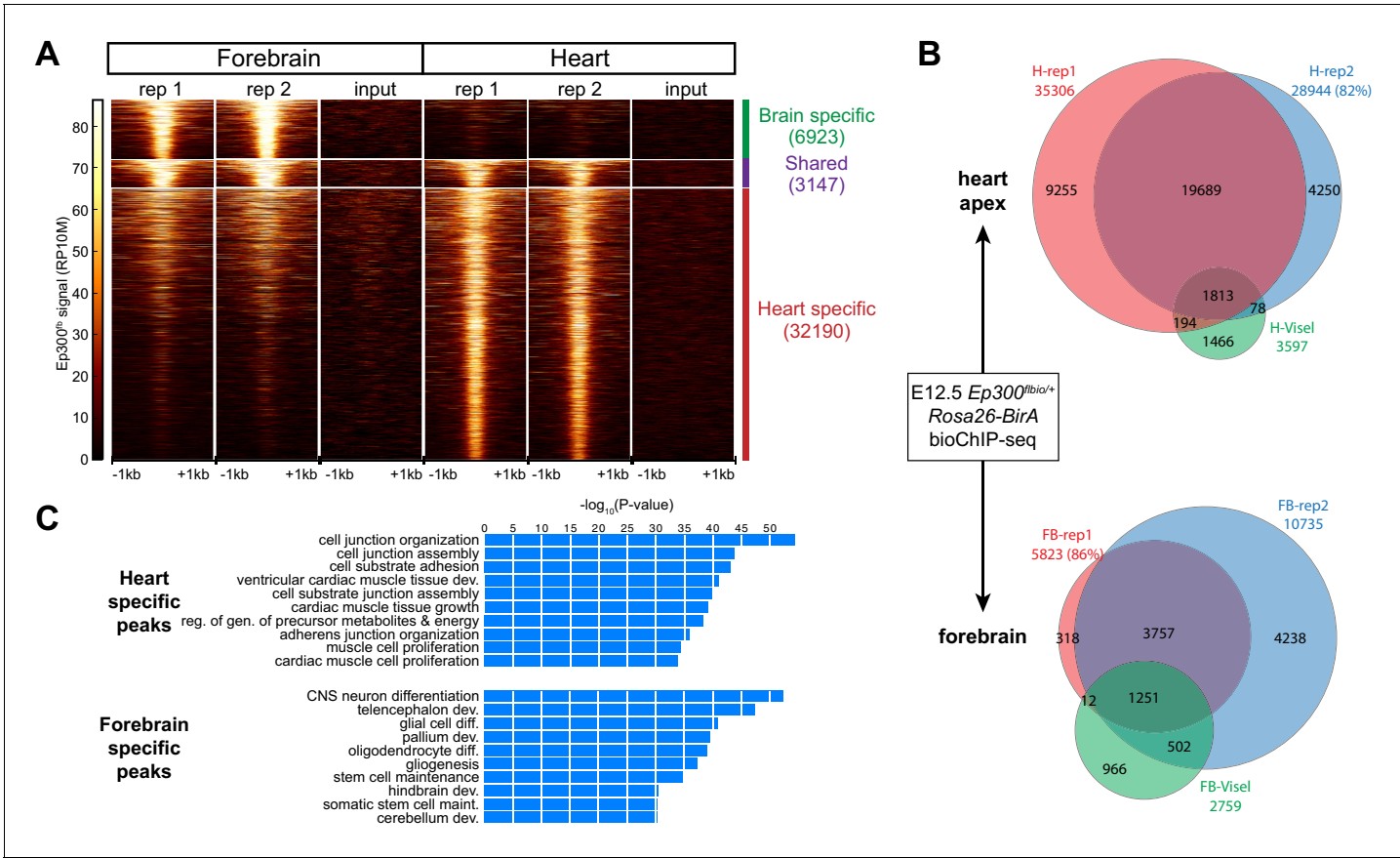

**Figure 3.** Tissue specific Ep300 bioChiP-seq. (A) Tag heatmap showing Ep300[fb] signal in heart and forebrain. Each row represents a region that was bound by Ep300 in heart, forebrain, or both. A minority of Ep300-enriched regions were shared between tissues. (B) Ep300[fb] pull down from E12.5 heart or forebrain. Ep300-bound regions identified by bioChIP-seq of *Ep300[fb/fb]; Rosa26[BirA]* tissues in biological duplicates are compared to regions identified by antibody-mediated Ep300 ChIP-seq (*Visel et al., 2009b*). (C) GO biological process terms enriched for genes that neighbor the tissue-specific heart or forebrain peaks. Bars indicate statistical significance.

The following figure supplements are available for figure 3:

**Figure supplement 1.** Heart and forebrain Ep300 bioChiP-seq.

**Figure supplement 2.** Comparison of heart Ep300[fb] regions to a compendium of heart enhancers assembled by *Dickel et al. (2016)*.

**Figure supplement 3.** Comparison of enhancer prediction based on indicated chromatin features of E12.5 heart.

We analyzed the prediction of active enhancers by our Ep300$^{fb}$ bioChiP-seq data. The VISTA Enhancer database (*Visel et al., 2007*) contains thousands of genomic regions that have been tested for tissue-specific enhancer activity using an in vivo transient transgenic assay. 185 tested regions had heart activity and 130 (70%) of these overlapped Ep300$^{fb}$ regions that were reproduced in both biological duplicates. In comparison, only 105 (57%) of these regions overlapped the regions previously reported to be bound by Ep300 using antibody ChIP-seq.

Recently human and mouse Ep300 and H3K27ac ChIP-seq data from fetal and adult heart were combined to yield a 'compendium' of heart enhancers, with the strength of ChIP-seq signal used to provide an 'enhancer score' ranging from 0 to 1 that correlated with the likelihood of regions covered in the VISTA database to show heart activity (*Dickel et al., 2016*). We compared our heart Ep300 regions to this compendium. Overall, 9438/72508 (13%) regions in the prenatal compendium overlapped with the Ep300 heart regions (*Figure 3—figure supplement 2A*). However, the overlap frequency increased markedly for regions with higher enhancer scores (*Figure 3—figure supplement 2B*). For example, if one considers the 3571 compendium regions with an enhancer score of at least 0.4 (corresponding to a validation rate in the VISTA database of ~25%), 2647 (74.1%) were contained within the heart Ep300 regions, and 63/68 (92.6%) regions with a score of at least 0.8 (validation rate ~43%) overlapped a heart Ep300 region. Thus, heart compendium regions that are more likely to have in vivo heart activity are largely covered by heart Ep300 regions. On the other hand, 10752 (53%) heart Ep300 regions were not covered by the compendium, suggesting that this database is incomplete, potentially as a result of its use of incomplete antibody-based Ep300 ChIP-seq data.

Ep300 antibody ChIP-seq was one of the criteria used to select some of the test regions in the VISTA Enhancer database; as an independent test free of this potentially confounding effect, we searched the literature for other heart enhancers that were confirmed using the transient transgenic assay. We identified 40 additional heart enhancers. Of these, 24 (60%) intersected the Ep300$^{fb}$ regions common to both replicates. In comparison, only 6/40 (15%) intersected the regions detected previously by Ep300 antibody ChIP-seq. Few heart enhancers were found in the regions unique to Ep300 antibody ChIP-seq (11/185 VISTA and 2/40 non-VISTA), compared to regions unique to Ep300$^{fb}$ bioChIP-seq (36/185 VISTA and 20/40 non-VISTA). We conclude that Ep300$^{fb}$ ChIP-seq predicts heart enhancers with sensitivity that is superior to antibody-mediated Ep300 ChIP-seq.

Other chromatin features have been used to predict transcriptional enhancers. We compared the accuracy of Ep300 bioChiP-seq to other chromatin features for heart enhancer prediction. To map accessible chromatin, we performed ATAC-seq (assay for transposable-accessible chromatin followed by sequencing) on E12.5 cardiomyocytes. E12.5 heart ChIP-seq data for modified histones (H3K27ac, H3K4me1, H3K4me2, H3K4me3, H3K9ac, H3K27me3, H3K9me3, H3K36me3) were obtained from publicly available datasets (see Materials and methods, Data Sources). Using a machine learning approach and the VISTA enhancer database as the gold standard, we evaluated the accuracy of each of these chromatin features, compared to Ep300 bioChiP-seq, for predictive heart enhancers (*Figure 3—figure supplement 3*). This analysis showed that Ep300 bioChiP-seq was the single most predictive chromatin feature (area under the receiver operating characteristic curve (AUC) = 0.805). ATAC-seq and H3K27ac also performed well (AUC = 0.749 and 0.747, respectively), whereas H3K4me1 had was poorly predictive (AUC = 0.589). Combining Ep300 bioChIP-seq with ATAC-seq improved predictive accuracy (AUC = 0.866), equivalent to the value obtained by performing predictions with all of the chromatin features (AUC = 0.862). These analyses indicate that of the features tested Ep300 is the best single factor for enhancer prediction.

## Cre-activated, lineage-specific Ep300$^{fb}$ bioChIP-seq

In vivo biotinylation of Ep300$^{fb}$ requires co-expression of the biotinylating enzyme BirA. We reasoned that Ep300$^{fb}$ bioChIP-seq could be targeted to a Cre-labeled lineage by making BirA expression Cre-dependent. Therefore, we established *Rosa26$^{fsBirA}$*, in which BirA expression is contingent upon Cre excision of a floxed-stop (fs) cassette (*Figure 4A*). In preliminary experiments, we showed that *Rosa26$^{fsBirA}$* expression of BirA was Cre-dependent (*Figure 4—figure supplement 1A*), as was Ep300$^{fb}$ biotinylation (*Figure 4B*). When activated by Cre driven from *Tek* regulatory elements (Tg (Tek-cre)1Ywa/J; also known as *Tie2Cre*), BirA was expressed in endothelial and blood lineages (*Figure 4C*), consistent with this Cre transgene's labeling pattern. Thus, *Rosa26$^{fsBirA}$* expresses BirA in a Cre-dependent manner.

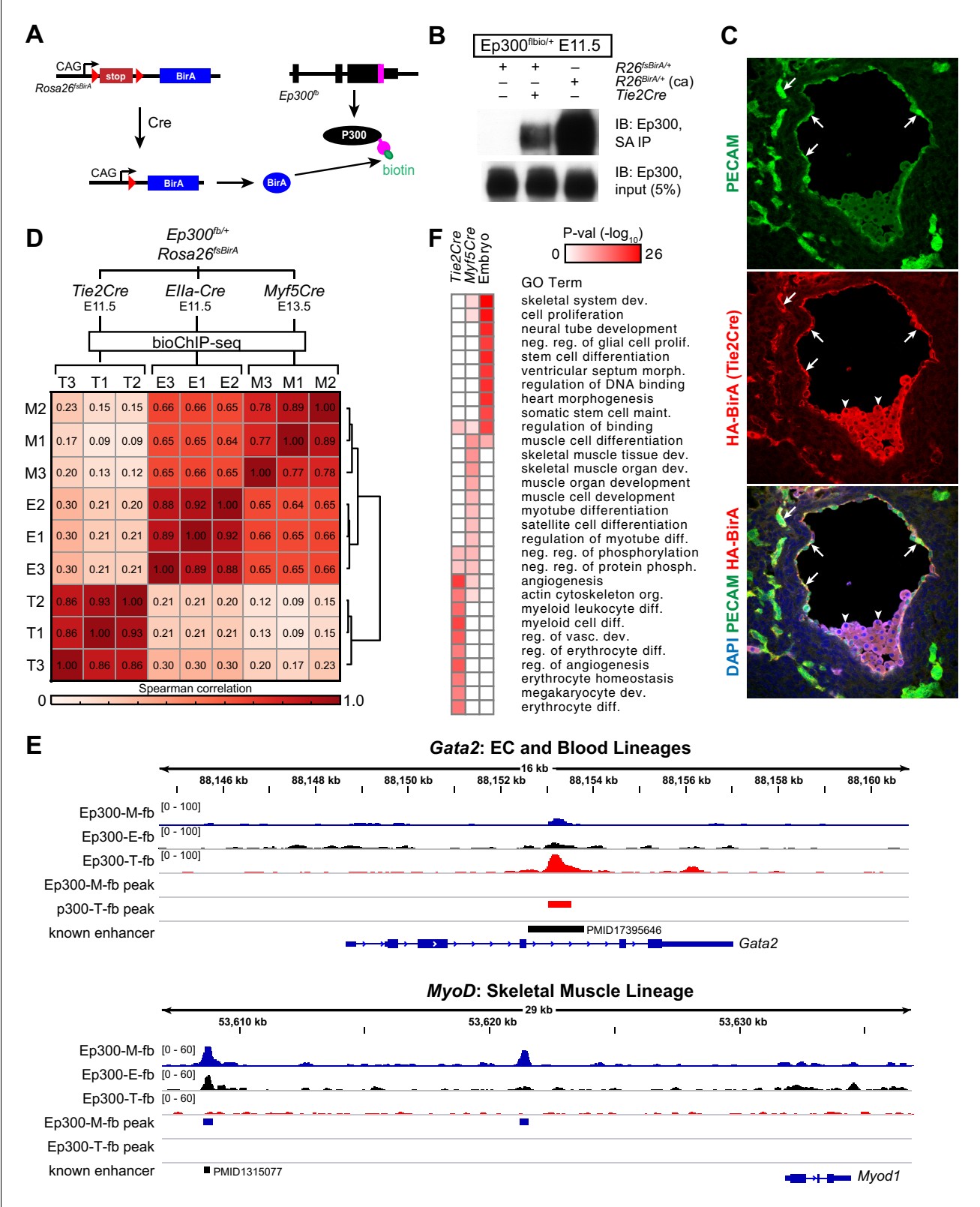

**Figure 4.** Cre-directed, lineage-selective Ep300 bioChiP-seq. (**A**) Experimental strategy. Lineage-specific Cre recombinase activates expression of BirA (HA-tagged) from Rosa26-flox-stop-BirA (*Rosa26^fsBirA*). This results in Ep300 biotinylation in the progeny of Cre-expressing cells. (**B**) Cre-dependent Ep300 biotinylation using *Rosa26^fsBirA*. Protein extracts were prepared from E11.5 embryos with the indicated genotypes. R26^BirA/+ (ca) and R26^fsBirA/+ indicate the constitutively active and Cre-activated alleles, respectively. (**C**) Immunostaining demonstrating selective *Tie2Cre*-mediated expression of

*Figure 4 continued*

HA-tagged BirA in ECs of R26$^{fsBirA}$; *Tie2Cre* embryos. Arrows and arrowheads indicate endothelial and hematopoietic lineages, respectively. (D) Tissue-selective Ep300 bioChIP-seq. *Tie2Cre* (T; endothelial and hematopoietic lineages), *Myf5Cre* (M; skeletal muscle lineages), and *EIIaCre* (E; germline activation) were used to drive tissue-selective Ep300 bioChIP-seq. Correlation between ChIP-seq signals within peak regions are shown for triplicate biological repeats. Samples within groups were the most closely correlated. (E) Ep300$^{fb}$ bioChIP-seq signal at *Gata2* (EC/blood specific) and *Myod* (muscle specific). Enhancers validated by transient transgenic assay are indicated along with the citation's Pubmed identifier (PMID). (F) Biological process GO terms illustrate distinct functional groups of genes that neighbor Ep300 bioChIP-seq driven by lineage-specific Cre alleles. The heatmap contains the top 10 terms enriched for genes neighboring each of the three lineage-selective Ep300 regions.

The following figure supplement is available for figure 4:

**Figure supplement 1.** Cre-dependent Ep300 bioChIP-seq.

Next, we compared Ep300$^{fb}$ bioChIP-seq from embryos when driven by *Tie2Cre* (endothelial and blood lineages) (*Kisanuki et al., 2001*), Myf5$^{tm3(cre)Sor}$/J (referred to as *Myf5Cre*; skeletal muscle lineage) (*Tallquist et al., 2000*), or Tg(EIIa-cre)C5379Lmgd/J (also known as *EIIaCre*; ubiquitous) (*Williams-Simons and Westphal, 1999*). For the *Tie2Cre* and *EIIaCre* samples, we used E11.5 embryos, a stage with robust angiogenesis. For *Myf5Cre*, we used E13.5 embryos, when muscle lineage cells are in a range of stages in the muscle differentiation program, spanning muscle progenitors to differentiated muscle fibers. BioChIP-seq from biological triplicates showed high within-group correlation, and lower between-group correlation, demonstrating the strong effect of different Cre transgenes in directing Ep300$^{fb}$ bioChIP-seq (*Figure 4D*). Viewing the bioChIP-seq signals in a genome browser confirmed lineage-selective signal enrichment. For example, *Tie2Cre* drove high Ep300$^{fb}$ bioChIP-seq signal at a *Gata2* intronic enhancer with known activity in endothelial and blood lineages (*Figure 4E*, top panel). There was less signal at this region in *Myf5Cre* and *EIIaCre* samples. At the skeletal muscle specific gene *Myod*, *Myf5Cre* drove strong Ep300$^{fb}$ bioChIP-seq signal at a known distal enhancer (*Goldhamer et al., 1992*), as well as a second Ep300 bound region about 12 kb upstream from the transcriptional start site.

To identify lineage-selective regions genome-wide, we filtered for regions with called peaks in which the lineage-specific Cre (*Tie2Cre* or *Myf5Cre*) Ep300$^{fb}$ signal was at least 1.5 times the ubiquitous Cre (*EIIaCre*) Ep300$^{fb}$ signal (*Figure 4—figure supplement 1B–C*). This led to the identification of 2411 regions with enriched signal in *Tie2Cre* (Ep300-T-fb) and 1292 regions with enriched signal in *Myf5Cre* (Ep300-M-fb), compared to 17382 regions with Ep300$^{fb}$ occupancy detected with ubiquitous biotinylation (Ep300-E-fb; *Supplementary file 1*). We analyzed the biological process gene ontology terms enriched for genes neighboring these three sets of Ep300 regions (*Figure 4F*). Ep300-T-fb regions were highly enriched for functional terms related to angiogenesis and hematopoiesis, whereas the Ep300-M-fb regions were highly enriched for functional terms related to skeletal muscle. Together, these results indicate that our strategy for Cre-driven, lineage-specific, Ep300$^{fb}$ bioChIP-seq successfully identifies regulatory regions that are associated with lineage relevant biological processes.

## Functional validation of enhancer activity of lineage-specific Ep300$^{fb}$ regions

We next set out to validate the in vivo enhancer activity of the regions with Cre-driven lineage-enriched Ep300$^{fb}$ occupancy. If a substantial fraction of the Ep300-T-fb regions have transcriptional enhancer activity, then genes neighboring these enhancers should be expressed at higher levels in *Tie2Cre* lineage cells. To test this hypothesis, we used *Tie2Cre*-activated translating ribosome affinity purification (*Heiman et al., 2008*; *Zhou et al., 2013*) (T2-TRAP) to obtain the gene expression of *Tie2Cre*-marked cell lineages. Using this lineage-specific expression profile, we then compared the expression of genes neighboring Ep300-T-fb, Ep300-M-fb, and Ep300-E-fb regions. Ep300-T-fb neighboring genes were more highly expressed compared to Ep300-E-fb neighboring genes ($p<10^{-38}$, Mann-Whitney U-test; *Figure 5A*). In contrast, there was no significant difference between Ep300-M-fb and Ep300-E-fb neighboring genes (*Figure 5A*). This result held regardless of the maximal distance threshold used to find the gene nearest to a Ep300 region (*Figure 5—figure supplement 1A*). We also compared the expression of genes with and without an associated Ep300-T-fb

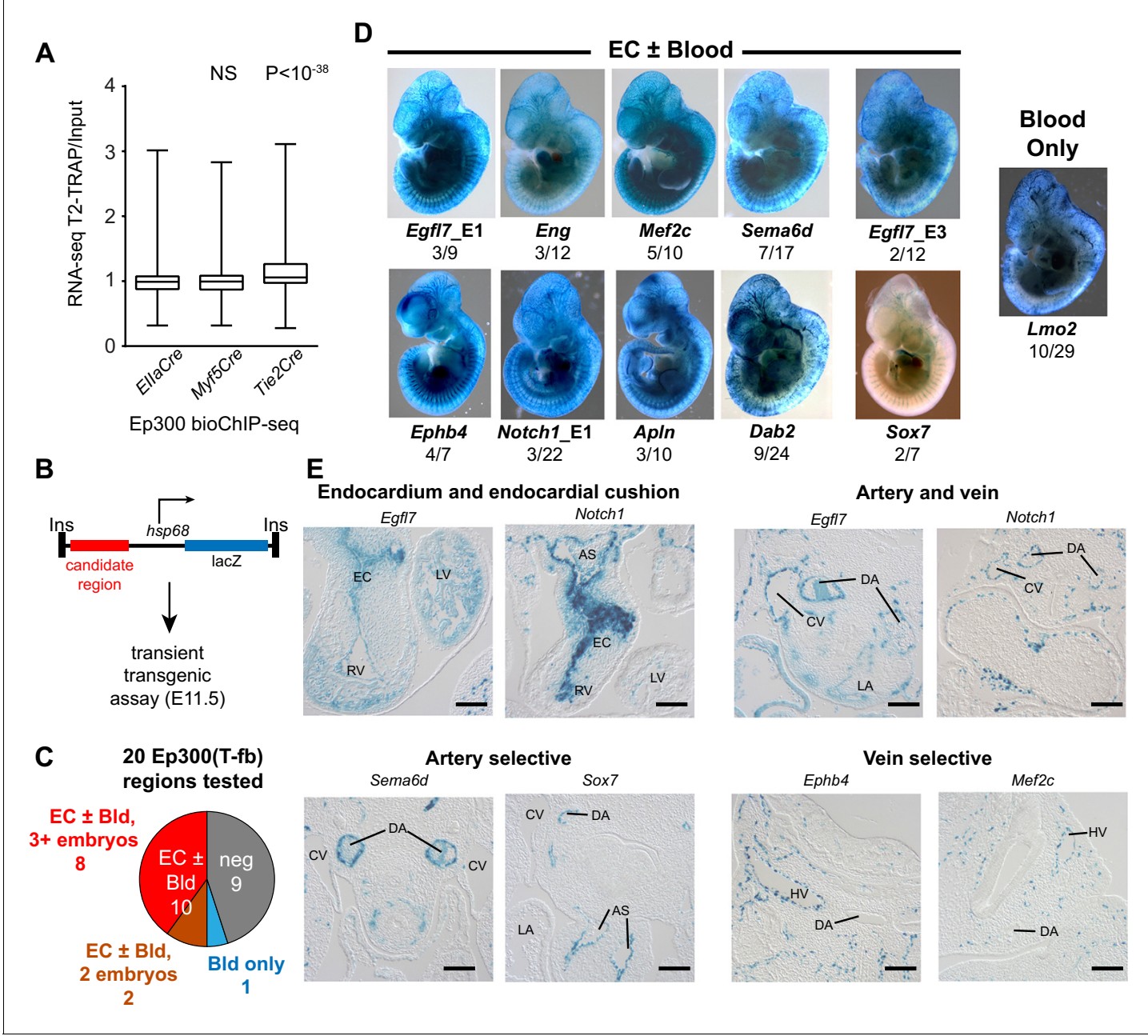

**Figure 5.** Functional validation of enhancer activity of Ep300-T-fb-bound regions. (**A**) Expression of genes neighboring regions bound by Ep300[fb] in different Cre-marked lineages. Translating ribosome affinity purification (TRAP) was used to enrich for RNAs from the *Tie2Cre* lineage. Input or *Tie2Cre*-enriched RNAs were profiled by RNA sequencing. The expression of the nearest gene neighboring Ep300 regions in ECs (Ep300-T-fb regions), but not skeletal muscle (Ep300-M-fb regions), was higher than that of genes neighboring regions bound by Ep300 across the whole embryo (*EIIaCre*). Box and whiskers show quartiles and 1.5 times the interquartile range. Groups were compared to *EIIaCre* using the Mann-Whitney U-test. (**B**) Transient transgenesis assay to measure in vivo activity of Ep300-T-fb regions. Test regions were positioned upstream of an *hsp68* minimal promoter and lacZ. Embryos were assayed at E11.5. (**C**) Summary of transient transgenic assay results. Out of 20 regions tested, nine showed activity in ECs or blood in three or more embryos, and two more showed activity in two embryos. See also *Table 2*. (**D**) Representative whole mount Xgal-stained embryos. Enhancers that directed LacZ expression in an EC or blood pattern in two or more embryos are shown. Numbers indicate embryos with LacZ distribution similar to shown image, compared to the total number of PCR positive embryos. (**E**) Sections of Xgal-stained embryos showing examples of enhancers active in arteries, veins, and endocardium, or selectively active in arteries or veins. AS: aortic sac; CV: cardinal vein; DA: dorsal aorta; EC: endocardial cushion; HV: head vein; LA: left atrium; LV: left ventricle; RV: right ventricle. Scale bars, 100 μm. See also *Figure 5—figure supplements 1* and *2* and *Table 2*.

*Figure 5 continued on next page*

*Figure 5 continued*

The following figure supplements are available for figure 5:

**Figure supplement 1.** Relationship of EC gene expression to Ep300 regions in ECs, skeletal muscle, and whole embryo.

**Figure supplement 2.** Transient transgenic assays to measure in vivo activity of candidate endothelial cell/blood enhancers.

**Figure supplement 3.** Histological sections of transient transgenic embryos.

region. Ep300-T-fb-associated genes were more highly expressed than non-associated genes (*Figure 5—figure supplement 1B*). Some Ep300-T-fb-associated genes were not detected within actively translating transcripts (*Figure 5—figure supplement 1C*). This suggests that in some cases Ep300 enhancer binding is not sufficient to drive gene expression; other contributing factors likely include imprecision of the enhancer-to-gene mapping rule, and regulation at the level of ribosome binding to transcripts. Together, our data are consistent with Ep300-T-fb regions being enriched for enhancers that are active in the *Tie2Cre*-labeled lineage.

To further evaluate the enhancer activity of Ep300-T-fb regions, we searched the literature and the VISTA Enhancer database (*Visel et al., 2007*) for genomic regions with endothelial cell activity validated by transient transgenesis (*Table 1*). Of 40 positive regions identified, 19 (47.5%) overlapped with Ep300-T-fb regions. Next, we used the transient transgenic assay to test the lineage-selective enhancer activity of 20 additional Ep300-T-fb regions. We selected regions that neighbored genes with EC-selective expression (T2-TRAP more than 10-fold enriched over input RNA) and with known or potential relevance to angiogenesis. Of the 20 tested Ep300-T-fb regions, eight drove reporter gene activity in at least three embryos in a vascular pattern, in both whole mounts and histological sections, and two additional regions drove reporter gene activity in a vascular pattern in two embryos (*Figure 5B–D*; *Table 2*; *Figure 5—figure supplements 2–3*). In retrospect, two of the positive enhancers (*Eng*; *Mef2c*) had been described previously (*Table 2*) (*Pimanda et al., 2006*; *De Val et al., 2008*). In some of these cases, there was also activity in blood cells, consistent with *Tie2Cre* activity in both blood and endothelial lineages. Additionally, one enhancer of *Lmo2* was active blood cells but not ECs. Thus a substantial fraction (9/20; 45%) of regions identified by lineage-selective, Cre-directed bioChIP-seq have appropriate and reproducible in vivo activity.

Arterial and venous ECs have overlapping but distinct gene expression programs, yet only three artery-specific and no vein-specific transcriptional enhancers have been described (*Wythe et al., 2013*; *Robinson et al., 2014*; *Becker et al., 2016*; *Sacilotto et al., 2013*). We examined histological sections of the transient transgenic embryos to determine if a subset is selectively active in ECs of the dorsal aorta or cardinal vein. We identified an enhancer, *Sema6d*-enh, with activity predominantly in ECs that line the dorsal aorta but not the cardinal vein (*Figure 5E*, *Table 2*). A second enhancer, *Sox7*-enh, also showed selective activity in the dorsal aorta, although this was only reproduced in two embryos. Both were also active in the endocardium and endocardial derivatives of the cardiac outflow tract (*Figure 5—figure supplement 3*). We also identified two enhancers with activity at E11.5 predominantly in ECs that line the cardinal vein and not the dorsal aorta (*Ephb4*-enh and *Mef2c*-enh; *Figure 5E*, *Table 2*, and *Figure 5—figure supplement 3*). Interestingly, a core 44-bp region of *Mef2c*-enh had been previously reported to drive pan-EC reporter expression at E8.5–E9.5 (*De Val et al., 2008*). This enhancer's activity pattern may be dynamically regulated at different developmental stages, as has been described previously for two artery-specific enhancers (*Robinson et al., 2014*; *Becker et al., 2016*). Further analysis of these enhancers will be required to confirm the artery- and vein- selective activity patterns that we observed, and to better characterize their temporospatial regulation.

Collectively, our data show that we have developed and validated a robust method for identification of transcriptional enhancers in Cre-marked lineages. Using this strategy, we discovered thousands of candidate skeletal muscle, EC, and blood cell enhancers. Based on our validation studies, we expect that a majority of these candidate regions have in vivo, cell-type specific transcriptional enhancer activity.

**Table 1.** mm9 genome coordinates of regions with EC activity as determined by transient transgenic assay. Vista_XXX indicates that the region was obtained from the VISTA enhancer database. Lifeover indicates that the region was inferred by liftover from the human genome. For enhancers obtained from the literature, Pubmed was searched for 'endothelial cell enhancer'. The resulting references were manually curated for transient transgenic testing of candidate endothelial cell enhancer regions.

| Chr | Start | End | Note |
|-----|-------|-----|------|
| chr9 | 37206631 | 37209631 | Robo4;PMID17495228;bloodVessels |
| chr4 | 94412022 | 94413640 | Tek;PMID9096345;bloodVessels |
| chr8 | 106625634 | 106626053 | Cdh5;PMID15746076;bloodVessels |
| chr11 | 49445663 | 49446520 | Flt4;posEC;liftoverFromPMID19070576;FoxETS |
| chr6 | 99338223 | 99339713 | FoxP1;posEC;liftoverFromPMID19070576;FoxETS |
| chr8 | 130910720 | 130911740 | Nrp1;posEC;liftoverFromPMID19070576;FoxETS |
| chr18 | 61219017 | 61219690 | Pdgfrb;posEC;liftoverFromPMID19070576;FoxETS |
| chr4 | 137475841 | 137476446 | Ece1;posEC;liftoverFromPMID19070576;FoxETS;artery |
| chr4 | 114743698 | 114748936 | Tal1;PMID14966269;endocardium;bloodVessels |
| chr2 | 119152861 | 119153661 | Dll4;PMID23830865;arterial |
| chr13 | 83721919 | 83721962 | Mef2c;PMID19070576;FoxETS |
| chr13 | 83711086 | 83711527 | Mefec;PMID15501228;panEC |
| chr2 | 32493213 | 32493467 | Eng;liftoverFromPMID16484587;bloodVessels |
| chr2 | 32517844 | 32518197 | Eng;liftoverFromPMID18805961;bloodVessels;blood |
| chr6 | 88152598 | 88153791 | Gata2;PMID17395646;PMID17347142;bloodVessels |
| chr9 | 32337302 | 32337549 | Fli1;PMID15649946;bloodVessels |
| chr5 | 76370571 | 76372841 | KDR;PMID10361126;bloodVessels |
| chr6 | 125502138 | 125502981 | VWF;PMID20980682;smallbloodVessels |
| chr5 | 148537311 | 148538294 | Flt1;liftoverFromPMID19822898 |
| chr17 | 34701455 | 34702277 | Notch4;liftoverFromPMID15684396 |
| chr2 | 155568649 | 155569132 | Procr;liftoverfromPMID16627757;bloodVessels[7/17] |
| chr9 | 63916890 | 63917347 | Smad6;liftoverfromPMID17213321;bloodVessels |
| chr19 | 37510161 | 37510394 | Hhex;liftoverfromPMID15649946;bloodVessels;blood |
| chr5 | 76357892 | 76358715 | Flk1;PMID27079877;bloodVessesl;artery |
| chr11 | 32145270 | 32146411 | vista_101;heart[9/12];bloodVessels[7/12] |
| chr13 | 28809515 | 28811310 | vista_265;neural tube[7/8];bloodVessels[3/8] |
| chr17 | 12982583 | 12985936 | vista_89;heart[6/10];bloodVessels[8/10] |
| chr4 | 131631431 | 131635142 | vista_80;bloodVessels[10/10] |
| chr4 | 57858433 | 57860639 | vista_261;bloodVessels[5/8] |
| chr5 | 93426293 | 93427320 | vista_397;limb[12/13];bloodVessels[12/13] |
| chr8 | 28216799 | 28218903 | vista_136;heart[7/10];other[6/10];bloodVessels[4/10] |
| chr3 | 87786601 | 87790798 | liftoverFromvista_1891;somite[7/7];midbrain;(mesencephalon)[6/7];limb[7/7];branchial;arch[7/7];eye[7/7];heart[7/7];ear[7/7];bloodVessels[6/7] |
| chr6 | 116309006 | 116309768 | liftoverFromvista_2065;bloodVessels[9/9] |
| chr19 | 37566512 | 37571839 | liftoverFromvista_1866;bloodVessels[5/5] |
| chr7 | 116256647 | 116260817 | liftoverFromvista_1859;neuraltube[8/8];hindbrain;(rhombencephalon)[5/8];midbrain;(mesencephalon)[8/8];forebrain[8/8];heart[7/8];bloodVessels[5/8];liver[4/8] |
| chr18 | 14006285 | 14007917 | liftoverFromvista_1653;bloodVessels[5/8] |
| chr2 | 152613921 | 152616703 | liftoverFromvista_2050;bloodVessels[5/5] |
| chr14 | 32045520 | 32047900 | liftoverFromvista_2179;bloodVessels[5/7] |
| chr15 | 73570041 | 73577576 | liftoverFromvista_1882;bloodVessels[8/8] |
| chr8 | 28216817 | 28218896 | liftoverFromvista_1665;heart[5/7];bloodVessels[7/7] |

**Table 2.** Summary of transient transgenic validation of candidate EC enhancers.

| Neighboring gene | Region (mm9) | Size (bp) | Location w/r gene | Distance to TSS | Whole mount | | | Sections | | | | Ref. (PMID) |
|---|---|---|---|---|---|---|---|---|---|---|---|---|
| | | | | | #PCR pos | #LacZ pos | # EC or blood pos | Endo | Art | Vein | Blood cells | |
| Apln | chrX:45358891–45359918 | 1028 | 3'_Distal | 28,624 | 10 | 3 | 3 | + | + | + | − | − |
| Dab2 | chr15:6009504–6010497 | 994 | 5'_Distal | −239,788 | 24 | 10 | 9 | + | ++ | + | +++ | − |
| Egfl7_enh1 | chr2:26427040–26428029 | 990 | 5'_Distal | −9,041 | 9 | 3 | 3 | +++ | +++ | +++ | + | − |
| Eng | chr2:32493216–32494019 | 804 | 5'_Distal | −8,497 | 12 | 4 | 3 | ++ | ++ | ++ | − | 16484587 |
| Ephb4 | chr5:137789649–137790412 | 764 | 5'_Proximal | −1,306 | 7 | 5 | 4 | ++ | − | +++ | − | − |
| Lmo2 | chr2:103733621–103734378 | 758 | 5'_Distal | −64,152 | 29 | 14 | 10 | − | − | − | +++ | − |
| Mef2c | chr13:83721522–83722451 | 930 | Intragenic | 78,954 | 10 | 6 | 5 | + | − | +++ | + | 19070576 |
| Notch1_enh1 | chr2:26330255–26331184 | 930 | Intragenic | −28,622 | 22 | 3 | 3 | +++ | ++ | ++ | − | − |
| Sema6d | chr2:124380522–124381285 | 764 | 5'_Distal | −55,128 | 17 | 10 | 7 | ++ | +++ | − | + | − |
| Egfl7_enh3 | chr2:26433680–26434642 | 963 | 5'_Distal | −2,415 | 12 | 2 | 2 | + | ++ | +++ | − | − |
| Sox7 | chr14:64576382–64577118 | 737 | 3'_Distal | 14,207 | 7 | 4 | 2 | + | ++ | − | − | − |
| Aplnr | chr2:85003436–85004412 | 977 | 3'_Distal | 27,407 | 12 | 3 | 0 | NA | NA | NA | NA | − |
| Egfl7_enh2 | chr2:26431273–26431995 | 723 | 5'_Proximal | −4,942 | 9 | 3 | 1 | + | − | − | − | − |
| Emcn | chr3:136984933–136986103 | 1171 | 5'_Distal | −18,524 | 15 | 1 | 1 | − | − | − | + | − |
| Ets1 | chr9:32481485–32482133 | 649 | Intragenic | −21,818 | 11 | 1 | 0 | NA | NA | NA | NA | − |
| Foxc1 | chr13:31921976–31922827 | 852 | 3'_Distal | 23,887 | 13 | 3 | 1* | NA | NA | NA | NA | − |
| Gata2 | chr6:88101907–88102696 | 790 | 5'_Distal | −46,356 | 8 | 0 | 0 | NA | NA | NA | NA | − |
| Lyve1 | chr7:118020264–118021043 | 780 | 5'_Distal | −3,128,028 | 7 | 1 | 1 | + | + | − | − | − |
| Notch1_enh2 | chr2:26345973–26347118 | 1146 | Intragenic | 12,796 | 9 | 0 | 0 | NA | NA | NA | NA | − |
| Sox18 | chr2:181397552–181398335 | 784 | 3'_Proximal | 8401 | 13 | 2 | 1 | − | ++ | − | − | − |

*EC/blood pattern on whole mount not validated in histological sections.

## Transcription factor binding motifs enriched in skeletal muscle and EC/blood enhancers

Ep300 does not bind directly to DNA. Rather, transcription factors recognize sequence motifs in DNA and subsequently recruit Ep300. The transcription factors and transcription factor combinations that direct enhancer activity in skeletal muscle, blood, and endothelial lineages are incompletely described. To gain more insights into this question, we searched for transcription factor binding motifs that were over-represented in the candidate enhancer regions bound by Ep300 in skeletal

muscle or blood/EC lineages. Starting from 1445 motifs for transcription factors or transcription factor heterodimers, we found 173 motifs over-represented in Ep300-T-fb or Ep300-M-fb regions (false discovery rate < 0.01% and frequency in Ep300 regions greater than 5%). Clustering and selection of representative non-redundant motifs left 40 motifs that were enriched in either Ep300-T-fb or Ep300-M-fb, or both (*Figure 6A*). Many closely related motifs were independently detected by de novo motif discovery (*Figure 6—figure supplement 1A*). Analysis of our T2-TRAP RNA-seq data identified genes expressed in embryonic ECs that potentially bind to these motifs (*Supplementary file 2*).

The GATA and ETS motifs were the most highly enriched motifs in the *Tie2Cre*-marked blood and EC lineages (*Figure 6A*). Interestingly, GO analysis of the subset of regions positive for these motifs showed that GATA-containing regions are highly enriched for hematopoiesis and heme synthesis, consistent with the critical roles of GATA1/2/3 in these processes (*Bresnick et al., 2012*). On the other hand, ETS-containing regions were highly enriched for functional terms linked to angiogenesis, also consistent with the key roles of ETS factors in angiogenesis (*Wei et al., 2009*) and our prior finding that the ETS motif is enriched in dynamic VEGF-dependent EC enhancers (*Zhang et al., 2013*). The Ebox motif, recognized by bHLH proteins, was the most highly enriched motif in the skeletal muscle lineage, consistent with the important roles of bHLH factors such as *Myod* and *Myf5* in skeletal muscle development (*Buckingham and Rigby, 2014*). Interestingly, the Ebox motif was also highly enriched in *Tie2Cre*-marked cells, and genes neighboring these Ebox-containing regions were functionally related to both angiogenesis and heme synthesis. bHLH-encoding genes such as *Hey1/2*, *Scl*, and *Myc* are known to be important in blood and vascular development.

The database used for our motif search included 315 heterodimer motifs that were recently discovered through high throughput sequencing of DNA concurrently bound by two different transcription factors (*Jolma et al., 2015*). This allowed us to probe for enrichment of heterodimer motifs that may contribute to enhancer activity in blood, EC, and skeletal muscle lineages. One heterodimer motif that was highly enriched in Ep300-T-fb regions (and not Ep300-M-fb regions; referred to as T2Cre-enriched motifs) was ETS:FOX2 (AAACAGGAA), comprised of a tail-to-tail fusion of Fox (TGTTT) and ETS (GGAA) binding sites (*Figure 6A*). GO analysis showed that this motif was closely linked to vascular biological process terms (*Figure 6A*). This motif was previously found to be sufficient to drive enhancer EC activity during vasculogenesis and developmental angiogenesis (*De Val et al., 2008*), validating that our approach is able to identify bona fide, functional heterodimer motifs. Interestingly, we discovered two additional ETS-FOX heterodimer motifs, which were also highly enriched in Ep300-T-fb regions and also linked to vascular biological process terms: ETS:FOX1 (GGATGTT), consisting of a head-to-tail fusion between ETS and FOX motifs, with the ETS motif located 5' to the FOX motif (*Figure 6A*, arrows over motif logo), and ETS:FOX3 (TGTTTACGGAA), a head-to-tail fusion with the FOX motif located 5' to the ETS motif. Other heterodimer motifs that were enriched in Ep300-T-fb regions and to our knowledge previously were unrecognized as regulatory elements in ECs were ETS:TBox, ETS:HOMEO, and ETS:Ebox. Similar analysis of Ep300-M-fb regions identified highly enriched Ebox-containing heterodimer motifs including Ebox:Hox, Ebox:HOMEO, and ETS:Ebox ('*Myf5Cre*-enriched motifs').

To assess functional significance of these motifs, we examined their evolutionary conservation. Using PhastCons genome conservation scores for 30 vertebrate species (*Siepel et al., 2005*), we measured the conservation of sequences matching *Tie2Cre*-enriched motifs within the central 200 bp of Ep300-T-fb regions. Whereas randomly selected 12 bp sequences from these regions exhibited a distribution of scores heavily weighted towards low conservation values, sequences matching *Tie2Cre*-enriched motifs showed a bimodal distribution consisting of highly conserved and poorly conserved sequences (*Figure 6B*). The conservation of individual heterodimer motifs such as ETS:FOX1-3 confirmed that they shared this bimodal distribution that included deeply conserved sequences (*Figure 6—figure supplement 1B*). These findings indicate that a subset of sequences matching *Tie2Cre*-enriched motifs, including the novel heterodimer motifs, are under selective pressure. *Myf5Cre*-enriched motifs within the center of Ep300-M-fb regions similarly adopted a bimodal distribution that includes a subset of motif occurrences with high conservation (*Figure 6B* and *Figure 6—figure supplement 1B*). This analysis supports the biological function of *Tie2Cre*- and *Myf5-Cre*-enriched motifs in endothelial cell/blood or skeletal muscle enhancers, respectively.

To further functionally validate the transcriptional activity of these heterodimer motifs, we measured their enhancer activity using luciferase reporter assays. Three repeats of enhancer fragments

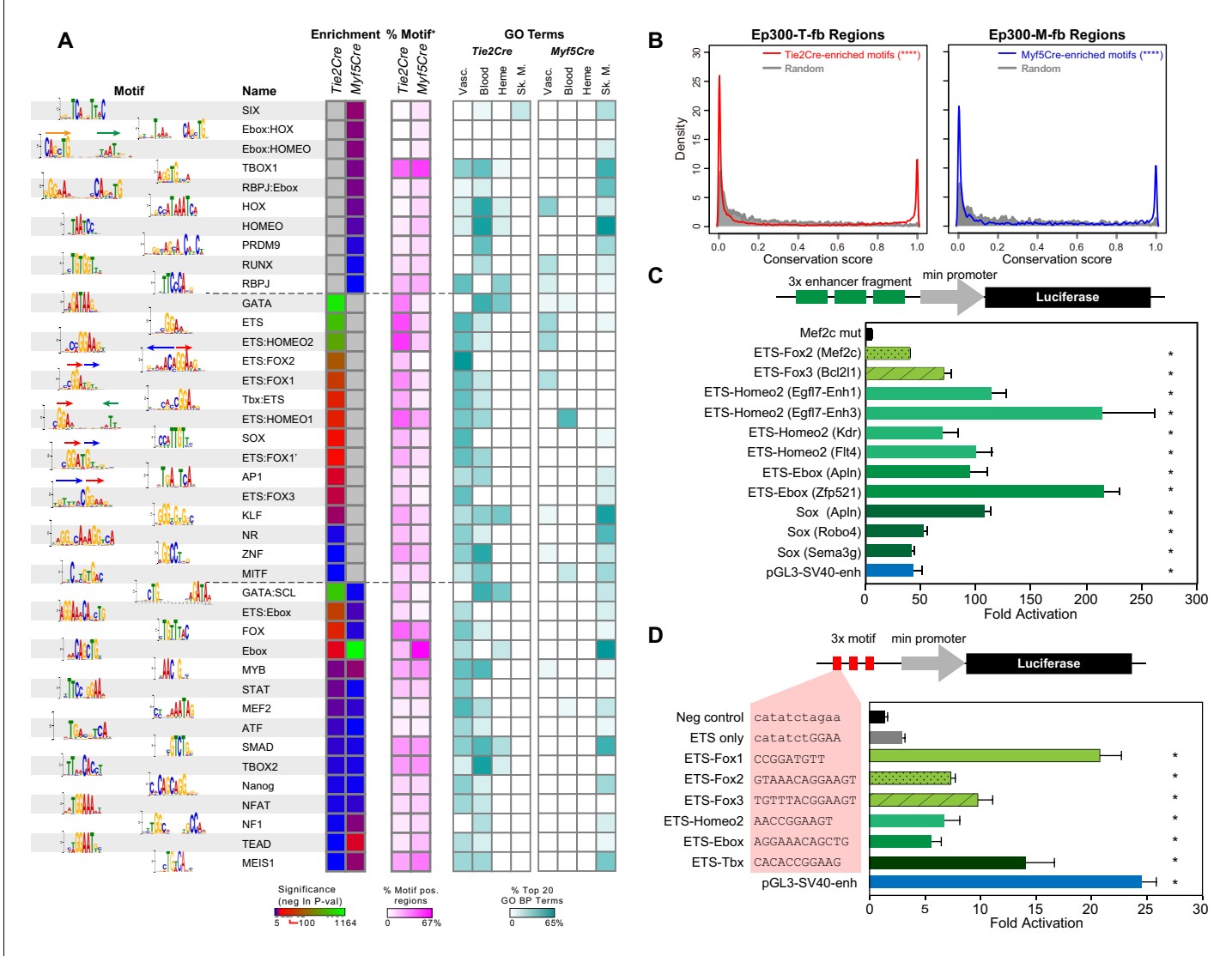

**Figure 6.** Motifs enriched in Ep300-T-fb and Ep300-M-fb regions. (A) Motifs enriched in Ep300-T-fb or Ep300-M-fb regions. 1445 motifs were tested for enrichment in Ep300 bound regions compared to randomly permuted control regions. Significantly enriched motifs (neg ln p-value>15) were clustered and the displayed non-redundant motifs were manually selected. Heatmaps show statistical enrichment (left), fraction of regions that contain the motif (center), and GO terms associated with genes neighboring motif-containing, Ep300-bound regions (fraction of the top 20 GO biological process terms). Grey indicates that the motif was not significantly enriched (neg ln p-value≤15). (B) Conservation of sequences matching *Tie2Cre-* or *Myf5Cre-*enriched motifs within 100 bp of the summit of Ep300-T-fb or Ep300-M-fb regions, compared to randomly selected 12 bp sequences from the same regions. PhastCons conservation scores across 30 vertebrate species were used. ****p<0.0001, Kolmogorov–Smirnov test. (C) Luciferase assay of activity of enhancers containing indicated motifs. Three repeats of 20–30 bp regions from Ep300-bound enhancers linked to the indicated gene and centered on the indicated motif were cloned upstream of a minimal promoter and luciferase. The constructs were transfected into human umbilical vein endothelial cells. Luciferase activity was expressed as fold activation above that driven by the enhancerless, minimal promoter-luciferase construct. *p<0.05 compared to Mef2c-enhancer with mutated ETS:FOX2 motif (Mef2c mut). n = 3. (D) Luciferase assay of indicated motifs repeated three times within a consistent DNA context. Assay was performed as in C. *p<0.05 compared to negative control sequencing lacking predicted motif. n = 3. Error bars in C and D indicate standard error of the mean.

The following figure supplement is available for figure 6:

**Figure supplement 1.** Motifs with enriched Ep300 occupancy in Tie2Cre or *Myf5Cre embryo samples.*

containing motifs of interest were cloned upstream of a minimal promoter and luciferase. The constructs were transfected into human umbilical vein endothelial cells (HUVECs), and transcriptional activity was measured by luciferase assay, normalized to the enhancerless promoter-luciferase construct (*Figure 6C*). The well-studied ETS:FOX2 motif from an endothelial *Mef2c* enhancer (*De Val et al., 2008*) robustly stimulated transcription to about the same extent as the SV40 enhancer. As expected, mutation of the ETS:FOX2 motif markedly blunted its activity, supporting the specificity of this assay. Interestingly, the alternative ETS:FOX3 motif uncovered by our study was at least as potent in stimulating transcription as the previously described ETS:FOX2 motif. The other novel heterodimer motifs tested, ETS:HOMEO2 and ETS:Ebox, likewise supported strong transcriptional activity, as did the SOX motif, whose enrichment and associated GO terms were highly EC-selective.

To further assess and compare the transcriptional activity of these heterodimer motifs, we cloned upstream of luciferase 3x repeated motifs into a consistent DNA context that had minimal endogenous enhancer activity (*Figure 6D*). The ETS motif alone only weakly stimulated luciferase expression, whereas the previously described ETS:FOX2 motif robustly activated luciferase expression. Interestingly, both of the newly identified, alternative ETS:FOX motifs (ETS:FOX1 and ETS:FOX3) were more potent activators that ETS:FOX2. The other heterodimer motifs tested, ETS-Homeo2, ETS-Ebox, and ETS-Tbox, also demonstrated significant enhancer activity.

Together, unbiased discovery of cell type specific enhancers coupled with motif analysis identified novel transcription factor signatures that are likely important for gene expression programs of blood, vasculature, and skeletal muscle.

## Organ-specific EC enhancers

ECs in different adult organs have distinct gene expression programs that underlie organ-specific EC functions (*Nolan et al., 2013*; *Coppiello et al., 2015*). For example, heart ECs are adapted for the transport of fatty acids essential for fueling oxidative phosphorylation in the heart (*Coppiello et al., 2015*). On the other hand, lung ECs are adapted for efficient transport of gas, but possess specialized tight junctions to minimize transit of water (*Mehta et al., 2014*). Nolan et al. recently profiled gene expression in ECs freshly isolated from nine different adult mouse organs (*Nolan et al., 2013*). Clustering the 3104 genes with greater than 4-fold difference in expression across this panel identified genes with selective expression in ECs from a subset of organs (*Figure 7—figure supplement 1A*). 240 genes were preferentially expressed in ECs from heart (and skeletal muscle) compared to other organs, whereas 355 genes were preferentially expressed in ECs from lung (*Figure 7—figure supplement 1B*). One cluster contained genes co-enriched in lung and brain, including many tight junction genes such as claudin 5.

We asked if our strategy of lineage-specific Ep300$^{fb}$ bioChIP-seq would allow us to identify enhancers linked to organ-specific EC gene expression. For these experiments, we used *Tg (Cdh5-cre/ERT2)1Rha* (also known as *VECad-CreERT2*) (*Sörensen et al., 2009*) to drive Ep300$^{fb}$ biotinylation in ECs (*Figure 7—figure supplement 2*); unlike *Tie2Cre*, this transgene does not label hematopoietic cells when induced with tamoxifen in the neonatal period. We isolated adult (eight wk old) heart and lungs from *Ep300$^{fb/+}$; VECad-CreERT2$^+$; Rosa26$^{fsBirA/+}$* mice and performed bioChIP-seq. Triplicate repeat experiments were highly reproducible (*Figure 7A*). Inspection of Ep300$^{fb}$ ChIP-seq signals from lung and heart ECs suggested that *VECad-CreERT2* successfully directed Ep300$^{fb}$ enrichment from ECs. For example, *Tbx3* and *Meox2*, transcription factors selectively expressed in lung and heart ECs, respectively, were associated with Ep300$^{fb}$-decorated regions in the matching tissues (*Figure 7—figure supplement 3*). Interestingly, *Meox2* has been implicated in directing expression of heart EC-specific genes and in the pathogenesis of coronary artery disease (*Coppiello et al., 2015*; *Yang et al., 2015*). On the other hand, *Tbx2/4* and *Myh6*, genes expressed in non-ECs in lung and heart, were not associated with regions of Ep300$^{fb}$ enrichment (*Figure 7—figure supplement 3*).

To obtain a broader, unbiased view of the adult EC Ep300$^{fb}$ bioChIP-seq results, regions bound by Ep300$^{fb}$ in either heart (14251) or lung (22174) were rank-ordered by the heart to lung Ep300$^{fb}$ signal ratio (*Supplementary file 1*). Regions were grouped into deciles by this ratio, with the most heart-enriched regions in decile one and the most lung-enriched regions in decile 10. Next, for genes neighboring regions in each decile, we compared expression in heart compared to lung. Genes neighboring decile one regions (greater Ep300$^{fb}$ signal in heart) had higher median mRNA transcript levels than lung (*Figure 7B*). In contrast, genes neighboring decile 10 regions (greater

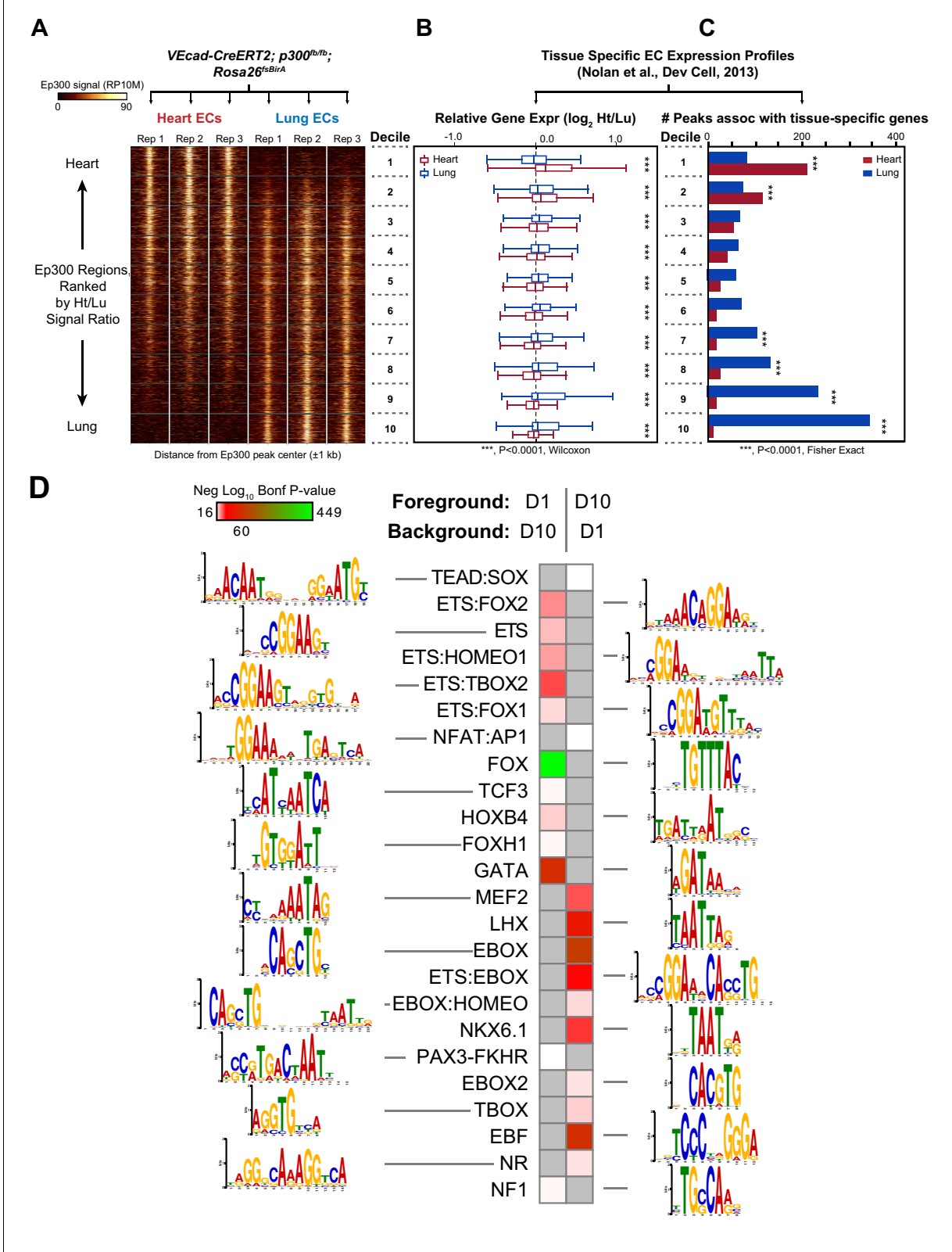

**Figure 7.** Enhancers in adult heart and lung ECs. *VEcad-CreERT2* and neonatal tamoxifen pulse was used to drive BirA expression in ECs. Ep300 regions in adult heart or lung ECs were identified by bioChIP-seq in biological triplicate. Ep300 regions were ranked into deciles by the ratio of the Ep300-VE-fb signal in heart to lung. (A) Tag heatmap shows that the top and bottom deciles have selective Ep300 occupancy in heart and lung ECs, respectively. (B) Expression of genes in heart ECs (red) or lung ECs (blue) neighboring Ep300 regions, divided into deciles by Ep300-VE-fb signal ratio in

*Figure 7 continued on next page*

*Figure 7 continued*

heart and lung. ***p<0.0001, Wilcoxon test. Expression values were obtained from (**Nolan et al., 2013**). Box plots indicate quartiles, and whiskers indicate 1.5 times the interquartile range. (C) Genes with selective expression in heart or lung ECs were identified by K-means clustering (see Materials and methods and *Figure 7—figure supplement 1*). The number of Ep300-VE-fb regions in heart (red) or lung (blue) neighboring these genes was determined, stratified by decile. ***p<0.0001, Fisher's exact test. (D) Enrichment of selected motifs in Ep300-VE-fb regions from heart (decile 1) compared to lung (decile 10) or vice versa. Grey indicates no significant enrichment. Displayed non-redundant motifs were selected from all significant motifs by manual curation of clustered motifs.

The following figure supplements are available for figure 7:

**Figure supplement 1.** Organ-specific EC gene expression.

**Figure supplement 2.** VEcad-CreERT2 activation of Ep300 biotinylation.

**Figure supplement 3.** Ep300$^{fb}$ bioChIP-seq signal in whole organ or ECs.

**Figure supplement 4.** Gene functional terms enriched in decile 1 or decile 10 of Ep300-VE-fb regions.

Ep300$^{fb}$ signal in lungs) had higher median mRNA transcript levels in lung than heart. We then focused on genes with selective expression in heart or lung. More decile 1 Ep300$^{fb}$ regions were associated with genes with heart-selective expression than lung-selective expression, and more decile 10 Ep300$^{fb}$ regions were associated with genes with lung-selective expression (in each case, p<0.0001, Fisher's exact test; *Figure 7C*). These results are consistent with greater heart- or lung-associated enhancer activity in Ep300$^{fb}$ decile 1 or 10 regions, respectively.

We analyzed the GO terms that are over-represented for genes neighboring Ep300$^{fb}$ decile 1 or 10 regions. Both sets of regions were highly enriched for GO terms related to vasculature and blood vessel development (*Figure 7—figure supplement 4*). Analysis of disease ontology terms showed that genes neighboring decile one regions (greater Ep300$^{fb}$ signal in heart) were significantly associated with terms relevant to coronary artery disease, such as 'coronary heart disease', 'arteriosclerosis', and 'myocardial infarction', whereas genes neighboring decile 10 regions (greater Ep300$^{fb}$ signal in lung) were less enriched. Conversely, genes neighboring decile 10 regions were selectively associated with hypertension and cerebral artery disease.

To gain insights into transcriptional regulators that preferentially drive heart or lung EC enhancers, we searched for motifs with differential enrichment in decile one compared to decile 10. Using decile one regions as the foreground sequences and decile 10 regions as the background sequences, we detected significant enrichment of the TCF motif in regions preferentially occupied by Ep300 in heart ECs (*Figure 7D*), consistent with prior work that found that a Meox2:TCF15 complex promotes heart EC-specific gene expression (*Coppiello et al., 2015*). Other motifs that were significantly enriched in decile 1 (heart) compared to decile 10 regions were FOX, ETS, ETS-FOX and ETS-HOMEO motifs. We performed the reciprocal analysis to identify lung-enriched motifs. One motif enriched in decile 10 (lung) compared to decile one regions was the TBOX motif. Interestingly, TBX3 is a lung EC-enriched transcription factor (*Nolan et al., 2013*). Other motifs over-represented in decile 10 compared to decile one regions were EBox, ETS:Ebox, and EBOX:HOMEO motifs. Thus differences in transcription factor expression or heterodimer formation in heart and lung ECs may contribute to differences in Ep300 chromatin occupancy and enhancer activity between heart and lung.

## Discussion

Here we show that Cre-mediated, tissue-specific activation of BirA permits high affinity bioChiP-seq of factors with a *bio* epitope tag. Furthermore, we show that combining this strategy with the Ep300$^{fb}$ knockin allele permits efficient identification of cell type-specific enhancers. The *bio* epitope has been knocked into a number of different transcription factor loci (*He et al., 2012*; *Waldron et al., 2016*) and Jackson Labs 025982 (*Rbpj*), 025980 (*Ep300*), 025983 (*Mef2c*), 025978 (*Nkx2-5*), 025979 (*Srf*), and 025977 (*Zfpm2*)), and these alleles can be combined with Cre-activated

BirA to permit lineage-specific mapping of transcription factor binding sites. Cell type-specific protein biotinylation will also be useful for mapping protein-protein interactions in specific cell lineages.

Using this technique, we identified thousands of candidate skeletal muscle and EC enhancers and showed that many of these candidate enhancers are likely to be functional. Furthermore, we showed that the technique can identify tissue-specific enhancers in postnatal tissues, and identified novel candidate enhancers that regulate organ-specific endothelial gene expression. These enhancer regions will be a valuable resource for future studies of transcriptional regulation in these systems.

Large scale identification of tissue-specific enhancers will facilitate decoding the mechanisms responsible for cell type-specific gene expression in development and disease. Here. By analyzing these candidate regulatory regions, we revealed novel transcriptional regulatory motifs that likely participate in skeletal muscle development, angiogenesis, and organ specific EC gene expression. Our recovery of a previously described FOX-ETS heterodimer binding site in EC enhancers (*De Val et al., 2008*) validates the ability of our approach to detect sequence motifs important for angiogenesis, and suggests that these motifs provide important clues to the transcriptional regulators that interact with them. Our study identified significant enrichment of two new FOX-ETS heterodimer motifs in which the FOX and ETS sites are in different positions and orientations. In their original study, Jolma and colleagues already demonstrated that these alternative FOX-ETS motifs are indeed bound by both FOX and ETS family proteins, implying that these proteins are able to collaboratively bind DNA in diverse configurations, potentially with DNA itself playing an important role in stabilizing the heterodimer (*Jolma et al., 2015*). Enrichment of these motifs in Ep300-T-fb, and their over-representation neighboring genes related to angiogenesis, suggests that these alternative FOX-ETS configurations are functional. We also recovered FOX-ETS motifs from Ep300 regions in adult ECs (and preferentially in heart ECs), suggesting that these motifs continue to be important in maintenance of adult vasculature. However, direct support of these inferences will require further experiments that identify the specific proteins involved and that dissect their functional roles in vivo.

Additional novel heterodimer motifs, such as ETS:EBox, ETS:HOMEO, and ETS:Tbox, were enriched in EC enhancers from both developing and adult ECs. This suggests that these motifs, and potentially the protein heterodimers that were reported to bind them (*Jolma et al., 2015*), are important for vessel growth and maintenance. These potential TF combinations, like the FOX-ETS combination, may act as a transcriptional code to create regulatory specificity at individual enhancers and their associated genes. Experimental validation of these hypotheses will be a fruitful direction for future studies.

### Limitations

A limitation of our current protocol is the need for several million cells to obtain robust ChIP-seq signal. Optimization of chromatin pulldown and purification through application of streamlined protocols or microfluidics (*Cao et al., 2015*), combined with the use of improved library preparation methods that work on smaller quantities of starting material, will likely overcome this limitation. Another limitation of our strategy is that Ep300 decorates many, but not all, active transcriptional enhancers (*He et al., 2011*), and therefore Ep300 bioChIP-seq will not comprehensively detect all enhancers. Our strategy does not directly permit profiling of other chromatin features in a Cre-targeted lineage. This limitation could be overcome by developing additional proteins labeled with the bio-epitope. For instance, by bio-tagging histone H3, non-tissue restricted ChIP for a feature of interest could be followed by sequential high affinity histone H3 bioChIP. Finally, our technique's lineage specificity is dependent on the properties of the Cre allele used (*Ma et al., 2008*), and users must be cognizant of the cell labeling pattern of the Cre allele that they choose.

## Materials and methods

### Mice

Animal experiments were performed under protocols approved by the Boston Children's Hospital Animal Care and Use Committee (protocols 13-08-2460R and 13-12-2601). *Ep300^{fb}* mice were generated by homologous recombination in embryonic stem cells. Targeted ESCs were used to generate a mouse line, which was bred to homozygosity. This line has been donated to Jackson labs (Jax 025980). The *Rosa26^{fsBirA}* allele was derived from the previously described *Rosa26^{fsTRAP}*

mouse (*Zhou et al., 2013*) (Jax 022367) by removal of the frt-TRAP-frt cassette using germline Flp recombination. The *Rosa26^BirA^* (*Driegen et al., 2005*) (constitutive; Jackson Labs 010920), *Tie2Cre* (*Kisanuki et al., 2001*) (Jackson Labs 008863), *Myf5Cre* (*Tallquist et al., 2000*) (Jackson Labs 007893), *EIIaCre* (*Williams-Simons and Westphal, 1999*) (Jackson Labs 003724), *Rosa26^mTmG^* (*Muzumdar et al., 2007*) (Jackson Labs 007576) and *VEcad-CreERT2* (*Sörensen et al., 2009*) (Taconic 13073) lines were described previously.

## Transient transgenics

Candidate regions approximately 1 kb in length were PCR amplified using primers listed in *Table 3* and cloned into a gateway-hsp68-lacZ construct derived from pWhere (Invivogen) (*He et al., 2011*). Constructs were linearized by PacI and injected into oocytes by Cyagen, Inc. At least 5 PCR positive embryos were obtained per construct. Regions were scored as positive for EC activity if they displayed an EC staining pattern in whole mount and validated in sections, using previously described criteria (*Visel et al., 2009a*). For scoring purposes, we required that an EC pattern was observed for at least three different embryos, although we also describe results for an additional two regions with activity observed in two different embryos. Embryos were analyzed at E11.5 by whole mount LacZ staining. After whole mount imaging, embryos were embedded in paraffin and sectioned for histological analysis.

## Histology

Embryos were collected in ice-cold PBS and fixed in 4% paraformaldehyde over night. For immunostaining, cryosections were stained with HA (Cell Signaling #3724, 1:100 dilution) and PECAM1 (BD Pharmingen, 553371, 1:200 dilution) antibodies and imaged by confocal microscopy (Olympus FV1000).

For immunostaining, cryosections were stained with HA (Cell Signaling #3724) and PECAM1 (BD Pharmingen, 553371) antibodies and imaged by confocal microscopy (Olympus FV1000).

## Ep300^fb/fb^;Rosa26^BirA/BirA^ ES cells derivation and bioChiP

ESCs were derived as described previously (*Bryja et al., 2006*). Five week old *Ep300^fb/fb^;Rosa26^BirA/BirA^* female mice were hormonally primed and mated overnight with eight weeks old *Ep300^fb/fb^; Rosa26^BirA/BirA^* male mice. Uteri were collected on 3.5 dpc and embryos were flushed out under a microscope. After removal of zona pellucida by treating the embryos with Tyrode's solution (Sigma, T1788), the embryos were cultured in ESC medium (DMEM with high glucose, 15% ES cell-qualified FBS, 1000 U/mL LIF, 100 μM non-essential amino acids, 1 mM sodium pyruvate, 2 mM glutamine,100 μM $\beta$-mercaptoethanol, and penicillin/streptomycin), supplemented with 50 μM PD98059 (Cell Signaling Technology, #9900) for 7–10 days. The outgrowth was dissociated with trypsin, and the cells were cultured in ESC medium in 24 well plates to obtain colonies, which were then clonally expanded. Five male ESC lines were retained for further experiments. Pluripotency of these ESC lines was confirmed by immunostaining for pluripotency markers Oct4, Sox2, and SSEA, and they were negative for mycoplasma.

The five *Ep300^fb/fb^;Rosa26^BirA/BirA^* ESC lines were cultured in 150 mm dishes to 70–80% confluence. Crosslinking was performed by adding formaldehyde to 1% and incubating at room temperature for 15 min. Chromatin was fragmented using a microtip sonicator (QSONICA Q700). The chromatin from 3 ESC lines was pooled for replicate one and from the other 2 ESC lines for replicate 2. Ep300^fb^ and bound chromatin were pulled down by incubation with streptavidin beads (Life Technologies #11206D).

## Cell culture and luciferase reporter assays

Candidate motif sequences were synthesized as oligonucleotides (*Table 3*) and cloned into plasmid pGL3-promoter (Promega) between MluI and XhoI. HUVEC cells (Lonza) were cultured to 50–60% confluent in 24-well plates and transfected in triplicate with 1 μg luciferase construct and 0.5 μg pRL-TK internal control plasmid, using 5 μl jetPEI-HUVEC (Polyplus). After 2 days, cells were analyzed using the dual luciferase assay (Promega). Luciferase activity was measured using a 96-plate reading luminometer (Victor2, Perkin Elmer). Results are representative of at least two independent experiments.

**Table 3.** Oligonucleotides used in this study.

**Genotyping primers**

| Name | Sequence (5'- > 3') | Comments |
|---|---|---|
| Ep300fb-f | AATGCTTTCACAGCTCGC | 0.28 kb for wild-type, 0.43 kb for Ep300fb knockin |
| Ep300fb-r | AAACCATAAATGGCTACTGC | |
| Forward common | CTCTGCTGCCTCCTGGCTTCT | Rosa26-fs-BirA, 0.33 kb for wildtype, 0.25 kb for knockin |
| Wild type reverse | CGAGGCGGATCACAAGCAATA | |
| CAG reverse | TCAATGGGCGGGGGTCGTT | |
| LacZ-f | CAATGCTGTCAGGTGCTCTCACTACC | 0.42 kb, genotyping of transient transgenic |
| LacZ-r | GCCACTTCTTGATGCTCCACTTGG | |

**Primers to amplify Ep300 peak regions for transient transgenic assay.**
4 nucleotides CACC have been added to all the forward primers for TOPO Cloning.

| Name | Sequence (5' - > 3') |
|---|---|
| Apln_f | CACCGGAGGCTGAGCAATGAATAG |
| Apln_r | TTGGCTGGGGAAGAGTAAGC |
| Aplnr_f | CACCTCTCTCTCTGGCTTCG |
| Aplnr_r | CCTCAGAATGTTTTCATGG |
| Dab2_f | CACCGTGGAAATCATAGCAC |
| Dab2_r | GGTTGGAATAAAAGAGC |
| Egfl7_Enh1_f | CACCGCCTACCCAGTGCTGTTCC |
| Egfl7_Enh1_r | CTGGAGTGGAGTGTCACG |
| Egfl7_Enh2_f | CACCGCTAGGGGCTTCTAGTTC |
| Egfl7_Enh2_r | AGGTCTCTTCTGTGTCG |
| Egfl7_Enh3_f | CACCTGTTAGTGGTGCTCCC |
| Egfl7_Enh3_r | TCCAAGGTCACAAAGC |
| Emcn_f | CACCAGCACACCTCGTAAAATGG |
| Emcn_r | GAGTGAAGTAAGACATCGTCC |
| Eng_f | CACCAAACTAATTAAAAAACAAAGCAGGT |
| Eng_r | CATATGTACATTAGAACCATCCA |
| Ephb4_f | CACCTGGGTCTCATCAACCGAAC |
| Ephb4_r | CCTATCTACATCAGGGCACTG |
| Ets1_f | CACCTTCGTCAGAAATGATCTTGCCA |
| Ets1_r | TAGCAAGAGAGCCTGGTCAG |
| Foxc1_f | CACCTCTCTGCTTCAAGGCACCTT |
| Foxc1_r | TGGATAGCATGCAGAGGACA |
| Gata2_f | CACCTTCTCTTGGGCCACACAGA |
| Gata2_r | ATCTGCTCCACTCTCCGTCA |
| Lmo2_f | CACCTGGTTTTGCTTGCTAC |
| Lmo2_r | CATTTCTAAGTCTCCAC |
| Lyve1_f | CACCTACTGCCATGGAGGACTG |
| Lyve1_r | AGACACCTGGCTGCCTGATA |
| Mef2c_f | CACCGGAGGATTAAAAATTCCCC |
| Mef2c_r | CCTCTTAAATGTACGTG |
| Notch1_Enh1_f | CACCTCCCAAATGCTCCACGATG |
| Notch1_Enh1_r | GAGGAATGGCGAGAAATAGAC |
| Notch1_Enh2_f | CACCGAAGGCAGGCAGGAATAAC |

*Table 3 continued on next page*

Zhou *et al.* eLife 2017;6:e22039. DOI: 10.7554/eLife.22039

*Table 3 continued*

**Genotyping primers**

| | |
|---|---|
| Notch1_Enh2_r | TGGACAGGTGCTTTGTTG |
| Sema6d_f | CACCTCTTAACCACTATCTCC |
| Sema6d_r | ACTTCCTACACAGTTC |
| Sox18_f | CACCTTGGGGGGAAAGAGTG |
| Sox18_r | GACTTCATCCCATCTC |
| Sox7_f | CACCACAGAGCCCCTGCATATGT |
| Sox7_r | GCATGGTTTCTGAAGCCCAAAT |

**3x repeated enhancer regions for luciferase assay.**
Core motifs of interest are highlighted in red.

| Name | Sequence (5'- > 3') |
|---|---|
| ETS-FOX2 (Mef2c) | CAGGAAGCACATTTGTCTACGCTTTCCTGTCATAACAGGAAGAGCAGGAA GCACATTTG TCTACGCTTTCCTGTCATAACAGGAAGAGCAGGAAGCAC ATTTGTCTACGCTTTCCTGTCATAACAGGAAGAG |
| Mef2c-mut | CAAGAAGCACATTTGTCTACGCTTTCCTGTCATATCTAGAAGAGCAAGAAGCACATTTG TCTACGCTTTCC TGTCATATCTAGAAGAGCAAGAAGCACATTTGTCTACGCTTTCCTGTCATATCTAGAAGAG |
| ETS-FOX3 (Bcl2l1) | CAGTTATTTCAGGAAAGATCAGTTATTTCAGGAAAGATCAGTTATTTCAGGAAAGAT |
| ETS-HOMEO2 (Egfl7_Enh1) | GACAGACAGGAAGGCGGGACAGACAGGAAGGCGGGACAGACAGGAAGGCGG |
| ETS-HOMEO2 (Egfl7_Enh3) | ACACACTTCCTGTTTCCTGACACACTTCCTGTTTCCTGACACACTTCCTGTTTCCTG |
| ETS-HOMEO2 (Flt4) | ACAGTCACTTCCTGTTTTACAGTCACTTCCTGTTTTACAGTCACTTCCTGTTTT |
| ETS-HOMEO2 (Kdr) | CAACAACAGGAAGTGGACAACAACAGGAAGTGGACAACAACAGGAAGTGGA |
| Sox (Apln) | CAGTTCCCCATTGTTCTCGCAGTTCCCCATTGTTCTCGCAGTTCCCCATTGTTCTCG |
| Sox (Robo4) | GCCAGAACAATGAAGAACAAAGCCTGCACGGCCAGAACAATGAAGAACAAAGCCTGCAC GGCCAGAACAATGAAGAACAAAGCCTGCACG |
| Sox (Sema3g) | CGAATGGAAAGGGCATTGTTCAGGGGAGAACGAATGGAAAGGGCATTGTTCAGGGGAGA ACGAATGGAAAGGGCATTGTTCAGGGGAGAA |
| ETS-Ebox (Apln) | AGGCGGAAGCAGCTGGGATAGGCGGAAGCAGCTGGGATAGGCGGAAGCAGCTGGGAT |
| ETS-Ebox (Zfp521) | TTATCCACAGGAAACAGATGAGGATCGTTATCCACAGGAAACAGATGAGGATCGTTATC CACAGGAAACAGATGAGGATCG |

**3x repeated motifs within a similar DNA context.**
Motifs are indicated in red.

| | |
|---|---|
| Neg. control | TGTCATATCTAGAAGAGTGTCATATCTAGAAGAGTGTCATATCTAGAAGAG |
| ETS_alone | TGTCATATCTGGAAGAGTGTCATATCTGGAAGAGTGTCATATCTGGAAGAG |
| ETS-FOX1 | TGTCCGGATGTTGAGTGTCCGGATGTTGAGTGTCCGGATGTTGAG |
| ETS-FOX2 | TGTGTAAACAGGAAGTGAGTGTGTAAACAGGAAGTGAGTGTGTAAACAGGAAGTGAG |
| ETS-FOX3 | TGTTGTTTACGGAAGTGAGTGTTGTTTACGGAAGTGAGTGTTGTTTACGGAAGTGAG |
| ETS_Ebox | TGTAGGAAACAGCTGGAGTGTAGGAAACAGCTGGAGTGTAGGAAACAGCTGGAG |
| ETS-HOMEO2 | TGTAACCGGAAGTGAGTGTAACCGGAAGTGAGTGTAACCGGAAGTGAG |
| Tbx-ETS | TGTCACACCGGAAGGAGTGTCACACCGGAAGGAGTGTCACACCGGAAGGAG |

## Western blotting

Immunoblotting was performed using standard protocols and the following primary antibodies: GAPDH, Fitzgerald, 10 R-G109A (1:10,000 dilution); Ep300, Millipore, 05257 (1:2000 dilution); BirA, Abcam, ab14002 (1:1000 dilution).

## Tissue collection for ChIP and bioChiP

bioChIP-seq was performed as described previously (*He et al., 2014*; *He and Pu, 2010*), with minor modifications. We used lower amplitude sonication to avoid fragmentation of Ep300 protein.

## Ep300 and H3K27ac in E12.5 heart and forebrain

E12.5 embryonic forebrain and ventricle apex tissues were isolated from Swiss Webster (Charles River) females crossed to *Ep300^{fb/fb};Rosa26^{BirA/BirA}* males. Cells were dissociated in a 2 mL glass dounce homogenizer (large clearance pestle, Sigma P0485) and then cross-linked in 1% formaldehyde-containing PBS for 15 min at room temperature. Glycine was added to final concentration of 125 mM to quench formaldehyde. Chromatin isolation was performed as previously described (*He and Pu, 2010*) 30 forebrains or 60 heart apexes were used in each sonication. Conditions were titrated to achieve sufficient fragmentation (mean fragment size 500 bp) while avoiding degradation of Ep300 protein. We used a microtip sonicator (QSONICA Q700) at 30% amplitude and a cycle of 5 s on and 20 s off for 96 cycles in total. Sheared chromation was precleared by incubation with 100 μl Dynabeads Protein A (Life Technologies, 10002D) for 1 hr at 4°C. For Ep300 bioChIP, 2/3 of the chromatin was then incubated with 250 μl (for forebrains) or 100 μl (for heart apexes) Dynabeads M-280 Streptavidin (Life Technologies, 11206D) for 1 hr at 4°C. The streptavidin beads were washed and bound DNA eluted as described (*He and Pu, 2010*). For H3K27ac ChIP, the remaining 1/3 chromatin was incubated with 10 μg (forebrain) or 5 μg (heart apex) H3K27ac antibody (ActiveMotif #39133) overnight at 4°C. Then 50 μl or 25 μl Dynabeads Protein A were added and incubated for 1 hr at 4°C. The magnetic beads were washed six times with RIPA buffer (50 mM HEPES, pH 8.0, 500 mM LiCl, 1% Igepal ca-630, 0.7% sodium deoxycholate, and 1 mM EDTA) and washed once in TE buffer. ChIP DNA was eluted at 65°C in elution buffer (10 mM Tris, pH 8.0, 1% SDS and 1 mM EDTA) and incubated at 65°C overnight to reverse crosslinks.

## Ep300 bioChIP of fetal EC cells

E11.5 embryos were isolated from pregnant *Rosa26^{mTmG}* females crossed to *Ep300^{fb/fb};Rosa26^{fsBirA/BriA};Tie2Cre* males. *Tie2Cre* positive embryos were picked under a fluorescence microscope for further experiments. Crosslinking and sonication were performed as described above. We used 15 *Tie2Cre* positive embryos for each bioChIP replicate. We used 750 μL Dynabeads M-280 Streptavidin beads for Ep300^{fb} pull-down; this amount was determined by empiric titration.

## Ep300 bioChIP of fetal Myf5Cre labeled cells

E13.5 embryos were isolated from pregnant *Rosa26^{mTmG}* females crossed to *Ep300^{fb/fb};Rosa26^{fsBirA/BriA};Myf5Cre* males. Cre positive embryos were picked under a fluorescence microscope for further experiments. five embryos were used for each bioChIP replicate and incubated with 750 μl Dynabeads M-280 Streptavidin beads for Ep300^{fb} pull-down.

## Ep300 bioChIP of adult ECs

*Ep300^{fb/fb};Rosa26^{fsBirA/mTmG};VEcad-CreERT2* pups were given two consecutive intragastic injections of 50 μl tamoxifen (2 mg/ml in sunflower seed oil) on postnatal day P1 and P2 to induce the activity of Cre. The lungs and heart apexes were collected when the mice were eight weeks old. Cross-linking and sonication were performed as described above. Six mice were used for each replicate. We used 600 μl (heart) or 1.5 mL (lung) streptavidin beads for the bioChIP.

## bioChIP-seq and ChIP-seq

Libraries were constructed using a ChIP-seq library preparation kit (KAPA Biosystems KK8500). 50 ng of sonicated chromatin without pull-down was used as input.

Sequencing (50 nt single end) was performed on an Illumina HiSeq 2500. Reads were aligned to mm9 using Bowtie2 (*Langmead and Salzberg, 2012*) using default parameters. Peaks were called with MACS2 (*Zhang et al., 2008*). Murine blacklist regions were masked out of peak lists. For embryo samples, peak calling was performed against input chromatin background with a false discovery rate of less than 0.01. For adult samples, peak calling was poor using input chromatin background and therefore was performed using the ChIP sample only at a false discovery rate of less than 0.05. Aggregation plots, tag heat maps, and global correlation analyses were performed using deepTools 2.0 (*Ramírez et al., 2014*). bioChIP-seq signal was visualized in the Integrated Genome Viewer (*Thorvaldsdóttir et al., 2013*).

To associate genomic regions to genes, we used Homer's AnnotatePeaks (*Heinz et al., 2010*) to select the gene with the closest transcriptional start site.

## ATAC-seq

E12.5 heart ventricles were dissociated into single cell-suspensions using the Neonatal Heart Dissociation Kit (Miltenyi Biotec #130-098-373) with minor modifications from the manufacturer's protocol. Embryonic tissue samples were incubated twice with enzyme dissociation mixes at 37°C for 15 min with gentle agitation by tube inversions between incubations. Cell mixtures were gently filtered through a 70 μm cell strainer and centrifuged at 300 x g, 5 min, 4°C. Red blood cells were lysed with 10X Red Blood Cell Lysis Solution (Miltenyi #130-094-183) and myocytes were isolated using the Neonatal Cardiomyocyte Isolation Kit (Miltenyi Biotec #130-100-825). 75,000 isolated cardiomyocytes were used for each ATAC-Seq experiment. Libraries were prepped as previously described (*Buenrostro et al., 2015*).

## TRAP

Translating ribosome RNA purification (TRAP) and RNA-seq were performed as described (*Zhou et al., 2013*). The expression of the gene with TSS closest to the Ep300-bound region was used to define 'neighboring gene'. In *Figure 5A*, no maximal distance limit was used; in *Figure 5— figure supplement 1*, a range of maximal distance thresholds were tested.

## Gene ontology analysis

Gene Ontology analysis was performed using GREAT (*McLean et al., 2010*). Results were ranked by the raw binomial P-value. To determine the fraction of terms relevant to a cell type or process, the top twenty biological process terms were manually inspected.

## Motif analysis

Homer (*Heinz et al., 2010*) was used for motif scanning and for de novo motif analysis. Regions analyzed were 100 bp regions centered on the summits called by MACS2. The motif database used for motif scanning was the default Homer motif vertebrate database plus the heterodimer motifs described by Jolma et al. (*Jolma et al., 2015*). To select motifs for display, motifs from samples under consideration with negative ln p-value > 15 in any one sample were clustered using STAMP (*Mahony et al., 2007*). Non-redundant motifs were then manually selected.

## Gene expression analysis

Translating ribosome RNA purification (TRAP) was performed as described (*Zhou et al., 2013*). E10.5 Rosa26$^{fsTrap/+}$;*Tie2Cre* embryos were isolated from Swiss Webster strain pregnant females crossed to Rosa26$^{fsTrap/Trap}$;*Tie2Cre* males. TRAP RNA from 50 embryos was pooled for RNA-seq. The polyadenylated RNA was purified by binding to oligo (dT) magnetic beads (ThermoFisher Scientific, 61005). RNA-seq libraries were prepared with ScriptSeq v2 kit (Illumina, SSV 21106) according to the manufacturer's instructions. RNA-seq reads were aligned with TopHat (*Trapnell et al., 2009*) and expression levels were determined with htseq-count (*Anders et al., 2015*). Adult organ EC expression values were obtained from *Nolan et al. (2013)* (GEO GSE47067).

## Comparison of enhancer prediction using different chromatin features

Enhancer regions in the VISTA database were used as the golden standard. Each chromatin feature that was analyzed was from E12.5 heart. Data sources are listed below under 'Data Sources'. For each chromatin factor, average read intensity in each VISTA regions with or without heart enhancer activity was calculated and used as the starting point for machine learning. We used the weighted KNN method as the classifier. The parameters used were: 10 nearest neighbors, Euclidean distance, and squared inverse weight. The classifier was trained and tested using 10-fold cross validation. ROC (Receiver operating characteristic) curve was used to evaluate the predictive accuracy of each factor, as measured by the area under the ROC curve (AUC).

## Conservation analysis

The phastCons score for the multiple alignments of 30 vertebrates to mouse mm9 genome was used as the conservation score (*Siepel et al., 2005*). Sequences within Ep300 regions matching selected motif (s) were identified using FIMO (*Grant et al., 2011*) with default settings, and the average conservation score across the width of the sequence was used. To generate the random conservation

background, 100 random motifs 12 bp wide (the average width of the motifs being analyzed) were used to the scan the same regions.

### Literature searches

Heart and endothelial cell enhancers were identified by searching PubMed for 'heart enhancer' or 'endothelial cell enhancer', respectively, and then manually curating references for mouse or human enhancers with appropriate activity in murine transient transgenic assays.

### Data sources

Sequencing data generated for this study are as follows: (1) ES_Ep300_bioChIP from $Ep300^{fb/fb}$; $Rosa26^{BirA/BirA}$ ESCs; (2) FB_Ep300_bioChIP from E12.5 $Ep300^{fb/+}$;$Rosa26^{BirA/+}$ forebrains; (3) FB_H3K27ac_ChIP from E12.5 $Ep300^{fb/+}$;$Rosa26^{BirA/+}$ forebrains; (4) He_Ep300_bioChIP from E12.5 $Ep300^{fb/+}$;$Rosa26^{BirA/+}$ heart apex; (5) He_H3K27ac_ChIP from $E12.5$ $Ep300^{fb/+}$;$Rosa26^{BirA/+}$ heart apex; (6) EIIaCre_Ep300_bioChIP from E11.5 $Ep300^{fb/+}$;$Rosa26^{BirA/+}$ whole embryos; Myf5-Cre_Ep300_bioChIP from E13.5 $Ep300^{fb/+}$;$Rosa26^{fsBirA/+}$;$Myf5Cre$ whole embryos; Tie2Cre_Ep300_-bioChIP from E11.5 $Ep300^{fb/+}$;$Rosa26^{fsBirA/+}$;$Tie2Cre^+$ whole embryos; Tie2Cre-TRAP and Tie2Cre-Input from E10.5 $Rosa26^{fsTrap/+}$;$Tie2Cre^+$ whole embryos; and ATAC-seq from wild-type E12.5 ventricular cardiomyocytes. These data are available via the Gene Expression Omnibus (accession number GSE88789) or the Cardiovascular Development Consortium server (https://b2b.hci.utah.edu/gnomex/; login as guest).

The following public data sources were used for this study: ES_Ep300_ChIP, GSE36027; ES_Ep300_input, GSE36027; E12.5_Histone_input, GSE82850, E12.5_H3K27ac, GSE82449; E12.5_H3K27me3, GSE82448; E12.5_H3K36me3, GSE82970; E12.5_H3K4me1, GSE82697; E12.5_H3K4me2, GSE82667; E12.5_H3K4me3, GSE82882; E12.5_H3K9ac, GSE83056; E12.5_H3K9me3, GSE82787. Antibody Ep300 ChIP-seq data on forebrain (*Visel et al., 2009b*) and heart (*Blow et al., 2010*) are from GSE13845 and GSE22549, respectively. Adult organ EC gene expression data are from GEO GSE47067 (*Nolan et al., 2013*).

### Accession numbers

Sequencing data generated for this study are available via the Gene Expression Omnibus (accession number GSE88789) or the Cardiovascular Development Consortium server (https://b2b.hci.utah.edu/gnomex/; login as guest; instructions for reviewer access are provided in a supplementary file).

## Acknowledgements

WTP was supported by funding from the National Heart, Lung, and Blood Institute (U01HL098166 and HL095712), by an Established Investigator Award from the American Heart Association, and by charitable donations from Dr. and Mrs Edwin A Boger. The content is solely the responsibility of the authors and does not necessarily represent the official views of the funding agencies.

## Additional information

### Funding

| Funder | Grant reference number | Author |
| --- | --- | --- |
| American Heart Association | 12EIA8440003 | William T Pu |
| National Institutes of Health | U01HL098166 | William T Pu |
| National Institutes of Health | U01HL095712 | William T Pu |

The funders had no role in study design, data collection and interpretation, or the decision to submit the work for publication.

### Author contributions

PZ, Conceived of the approach, Designed and performed experiments, Analyzed data, and co-wrote and Edited the manuscript; FG, Analyzed data and Edited the manuscript; LZ, Constructed plasmids

and Acquired data on transient transgenic embryos; BNA, Performed ATAC-seq; QM, Contributed to histological analysis of transient transgenic embryos; KL, ZL, Acquired data; AH, Provided reagents and Expertise on bioChIP-seq; SMS, Contributed to animal husbandry and Data acquisition; BZ, Generated the Ep300fb mice; WTP, Supervised the project, Designed experiments, Analyzed data, and co-wrote the manuscript

## Author ORCIDs
Brynn N Akerberg, http://orcid.org/0000-0001-6470-6588
Aibin He, http://orcid.org/0000-0002-3489-2305
William T Pu, http://orcid.org/0000-0002-4551-8079

## Ethics

Animal experimentation: Animal experiments were performed under protocols approved by the Boston Children's Hospital Animal Care and Use Committee (protocols 13-08-2460R and 13-12-2601).

## Additional files

### Supplementary files

• Supplementary file 1. Tissue-specific Ep300-bound regions identified in this study. Each tab of the excel spreadsheet contains the Ep300-bound regions in the following conditions: Each tab of this spreadsheet shows the Ep300-bound regions in the following samples. H1.narrowPeak: E12.5 heart, replicate 1. H2.narrowPeak: E12.5 heart, replicate 2. FB1.narrowPeak: E12.5 forebrain, replicate 1. FB2.narrowPeak: E12.5 forebrain, replicate 2. Ep300-T-fb: *Tie2Cre*-enriched regions. Average Ep300 signal in *Tie2Cre* (T2), *Myf5Cre* (M5), and *EIIaCre* (E) is shown in reads per million. T2/E and M5/E show the ratio of signals. *Tie2Cre*-enriched regions were defined as peak regions with T2/E ratio > 1.5. Ep300-M-fb: *Myf5Cre*-enriched regions. Average Ep300 signal in *Tie2Cre* (T2), *Myf5Cre* (M5), and *EIIaCre* (E) is shown in reads per million. T2/E and M5/E show the ratio of signals. *Myf5-Cre*-enriched regions were defined as peak regions with M5/E ratio > 1.5. Ep300-E-fb: Ep300-bound peaks called from whole embryo (ubiquitous BirA expression). Ep300-VE-fb: Merged *VEcad-CreERT2*-driven Ep300 peaks in adult heart and lung. The peaks were ranked by the ratio of Ep300 signal in heart ECs compared to lung ECs and then grouped into deciles.

• Supplementary file 2. Transcription factors expressed in embryonic ECs. E10.5 embryo T2-TRAP and input RNA-seq data were analyzed to identify DNA-binding transcriptional regulators with detectable expression level in ECs (T2-TRAP log2 (fpkm +1)>0.8) and preferential EC expression (T2-TRAP/input >1). Genes with DNA binding domains belonging to the indicated families were annotated based on literature searches and previously publshed catalogs of transcriptional regulators (*Fulton et al., 2009*; *Kanamori et al., 2004*).

### Major datasets

The following dataset was generated:

| Author(s) | Year | Dataset title | Dataset URL | Database, license, and accessibility information |
| --- | --- | --- | --- | --- |
| Zhou P, Gu F, Zhang L, Akerberg BN, Ma Q, Li K, He A, Lin Z, Stevens SM, Zhou B, Pu WT | 2016 | Mapping cell type-specific transcriptional enhancers using high affinity, lineage-specific p300 bioChIP-seq | https://www.ncbi.nlm.nih.gov/geo/query/acc.cgi?acc=GSE88789 | Publicly available at the NCBI Gene Expression Omnibus (accession no: GSE88789) |

The following previously published datasets were used:

| Author(s) | Year | Dataset title | Dataset URL | Database, license, and accessibility information |
| --- | --- | --- | --- | --- |
| Ren B, Shen Y | 2012 | Transcription Factor Binding Sites | https://www.ncbi.nlm. | Publicly available at |

| | | | | | |
|---|---|---|---|---|---|
| | | by ChIP-seq from ENCODE/LICR | | nih.gov/geo/query/acc.cgi?acc=GSE36027 | the NCBI Gene Expression Omnibus (accession no: GSE36027) |
| Bing Ren | 2016 | ChIP-seq from heart (ENCSR646GHA) | | https://www.ncbi.nlm.nih.gov/geo/query/acc.cgi?acc=GSE82850 | Publicly available at the NCBI Gene Expression Omnibus (accession no: GSE82850) |
| Bing Ren | 2016 | ChIP-seq from heart (ENCSR123MLY) | | https://www.ncbi.nlm.nih.gov/geo/query/acc.cgi?acc=GSM2191196 | Publicly available at the NCBI Gene Expression Omnibus (accession no: GSM2191196) |
| Visel A, Blow MJ, Pennacchio LA | 2009 | ChIP-seq Accurately Predicts Tissue-Specific Activity of Enhancers | | https://www.ncbi.nlm.nih.gov/geo/query/acc.cgi?acc=GSE13845 | Publicly available at the NCBI Gene Expression Omnibus (accession no: GSE13845) |

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
