## [Decision Letter]

Thank you for submitting your article "Mapping cell type-specific transcriptional enhancers using high affinity, lineage-specific p300 bioChIP-seq" for consideration by *eLife*. Your article has been reviewed by two peer reviewers, and the evaluation has been overseen by a Reviewing Editor and Kevin Struhl as the Senior Editor. The following individuals involved in review of your submission have agreed to reveal their identity: Benoit G Bruneau (Reviewer #1); Joshua D Wythe (Reviewer #2).

The reviewers have discussed the reviews with one another and the Reviewing Editor has drafted this decision to help you prepare a revised submission.

Summary:

We appreciate the novel in vivo biotinylation strategy, via generation of a flag-bio tagged allele of endogenous p300 in combination with a published R26-lsl-birA line and multiple tissue-specific Cre drivers, which demonstrated that p300 occupancy predicts lineage-specific transcription. Validation of the novel p300 biotinylation system in vivo through rigorous criteria demonstrate the utility of this reagent in predicting enhancer usage. We believe this is a high quality manuscript that will be of interest to the general community. However, there are a number of points that should be addressed, which are detailed below. Most importantly, please address the issue of how "neighboring genes" were determined and also describe the metrics by which enhancers were determined be positive in transgenic embryos.

Essential revisions:

1) The authors state, after performing the TRAP-seq, that they compared the expression in ECs of "neighboring genes". However, we could not find within the manuscript any bioinformatic criteria explaining how they decided what a neighboring gene was. The genomic criteria (distance +/- from the TSS, for example) for suggesting a relationship of p300 occupancy and translating ribosomal-bound message should be clearly stated (and references to other papers using the same criteria would be appropriate, as well as comparison to genes that are actively translated without p300 enrichment in these tissues). Additionally, they should discuss many p300-T-fb enriched regions were near genes that weren't actively translating in the EC sample.

2) Overall, the criteria of 2 embryos (out of at least 5 PCR positive embryos) leading to a call as a "positive" enhancer lacks any interpretation or comment on the strength of the enhancer, as some of their elements (like Sox7, for instance) appear very weak and others appear somewhat non-specific (like Apln in the AER of the forelimb and hindlimb, or Egfl7_3 – which doesn't appear to be endothelial enriched or specific). Additionally, scoring an enhancer as positive when only 3 out of 22 showed EC activity seems arbitrary (especially for Notch1, where it is not even obvious that the 3rd embryo shown has activity). The transgenic results are somewhat puzzling, as the Sox7 enhancer (Figure 5) appears to clearly have signal in the atria on the wholemount view, but there is no myocardial signal in the section. Were these from 2 different samples? Perhaps a more nuanced interpretation of the results, and maybe commenting on the strength of the enhancers, would be useful. Finally, it is unclear what is meant by stating that 55% activity is consistent with the validation rate of other reports (as forebrain was 87%, midbrain 88%, and limb 88% in Visel et al., 2009; Blow et al. ranged between ~65-75%, etc.).

---

## [Author Response]

*Essential revisions:*

*1) The authors state, after performing the TRAP-seq, that they compared the expression in ECs of "neighboring genes". However, we could not find within the manuscript any bioinformatic criteria explaining how they decided what a neighboring gene was. The genomic criteria (distance +/- from the TSS, for example) for suggesting a relationship of p300 occupancy and translating ribosomal-bound message should be clearly stated (and references to other papers using the same criteria would be appropriate, as well as comparison to genes that are actively translated without p300 enrichment in these tissues). Additionally, they should discuss many p300-T-fb enriched regions were near genes that weren't actively translating in the EC sample.*

The definition we used was the TSS closest to the p300 peak, without a maximal distance. This definition has been widely used, for example Blow et al., Nature Genetics, 2010. In the revision we also repeated the comparison with a series of different maximal distance thresholds and showed that the overall conclusion is not affected by the specific maximal threshold used (Figure 5—figure supplement 1).

As suggested, we also compared expression of p300-associated genes to those without a p300 peak. We found that p300-associated genes were more highly expressed.

Finally, we calculated the fraction of genes with and without associated p300 that were not detected within actively translating transcripts. We found that a higher fraction of p300-associated genes were detectably expressed. However, not all p300-associated genes were expressed, and not all expressed genes were p300-associated. This likely reflects multiple mechanisms of transcriptional activation, as well as imperfect rules used to associate genes to p300 regions.

*2) Overall, the criteria of 2 embryos (out of at least 5 PCR positive embryos) leading to a call as a "positive" enhancer lacks any interpretation or comment on the strength of the enhancer, as some of their elements (like Sox7, for instance) appear very weak and others appear somewhat non-specific (like Apln in the AER of the forelimb and hindlimb, or Egfl7_3 – which doesn't appear to be endothelial enriched or specific). Additionally, scoring an enhancer as positive when only 3 out of 22 showed EC activity seems arbitrary (especially for Notch1, where it is not even obvious that the 3rd embryo shown has activity). The transgenic results are somewhat puzzling, as the Sox7 enhancer (Figure 5) appears to clearly have signal in the atria on the wholemount view, but there is no myocardial signal in the section. Were these from 2 different samples? Perhaps a more nuanced interpretation of the results, and maybe commenting on the strength of the enhancers, would be useful. Finally, it is unclear what is meant by stating that 55% activity is consistent with the validation rate of other reports (as forebrain was 87%, midbrain 88%, and limb 88% in Visel et al., 2009; Blow et al. ranged between ~65-75%, etc.).*

We agree that scoring regions as positive or negatively overly simplifies the activity, which also includes spatial pattern, specificity, and strength. For this reason, we provide the raw data on the regions, as both whole embryo images and sections. However, scoring regions as positive or negative simplifies summarizing the activity of many regions, and comparison to prior studies.

We adopted the definition of an active enhancer that has been used by the Pennacchio and Visel groups. In their series of studies, an active enhancer was defined as one that drove expression in the target tissue (excluding embryos with ubiquitous expression, which likely reflect an integration site effect). In other words, a “heart enhancer” is an enhancer that drives expression in the heart, without regard to its specificity for heart. The Pennacchio/Visel definition of a positive enhancer is one that shows expected activity in three or more independent embryos. This demonstrates reproducibility. In our original manuscript, we used a criteria of two or more independent embryos, but in the revision we adhered to the definition of 3 or more for calculation of the validation rate. We retained the description of the results for the additional two enhancers with reproducible activity in two but not 3 embryos as these we feel it likely that this activity is biologically meaningful. We rewrote the text to reflect these changes and to remove the statement that 55% activity is consistent with previously reported validation rates.

For Sox7, the enhancer had activity in the endocardium of the outflow tract, not the atria. This is accurately reflected in the tissue sections.